# Extended Spectrum β-Lactamase-Producing Enterobacterales of Shrimp and Salmon Available for Purchase by Consumers in Canada—A Risk Profile Using the Codex Framework

**DOI:** 10.3390/antibiotics12091412

**Published:** 2023-09-06

**Authors:** F. Carl Uhland, Xian-Zhi Li, Michael R. Mulvey, Richard Reid-Smith, Lauren M. Sherk, Hilary Ziraldo, Grace Jin, Kaitlin M. Young, Mark Reist, Carolee A. Carson

**Affiliations:** 1Centre for Foodborne, Environmental and Zoonotic Infectious Diseases, Public Health Agency of Canada, Guelph, ON N1H 7M7, Canada; 2Veterinary Drugs Directorate, Health Products and Food Branch, Health Canada, Ottawa, ON K1A 0K9, Canada; 3National Microbiology Laboratory, Public Health Agency of Canada, Winnipeg, MB R3E 3R2, Canada

**Keywords:** risk profile, Codex, Enterobacterales, extended-spectrum β-lactamase, salmon, shrimp, seafood, whole genome sequencing

## Abstract

The extended-spectrum β-lactamase (ESBL)-producing Enterobacterales (ESBL-EB) encompass several important human pathogens and are found on the World Health Organization (WHO) priority pathogens list of antibiotic-resistant bacteria. They are a group of organisms which demonstrate resistance to third-generation cephalosporins (3GC) and their presence has been documented worldwide, including in aquaculture and the aquatic environment. This risk profile was developed following the Codex Guidelines for Risk Analysis of Foodborne Antimicrobial Resistance with the objectives of describing the current state of knowledge of ESBL-EB in relation to retail shrimp and salmon available to consumers in Canada, the primary aquacultured species consumed in Canada. The risk profile found that Enterobacterales and ESBL-EB have been found in multiple aquatic environments, as well as multiple host species and production levels. Although the information available did not permit the conclusion as to whether there is a human health risk related to ESBLs in Enterobacterales in salmon and shrimp available for consumption by Canadians, ESBL-EB in imported seafood available at the retail level in Canada have been found. Surveillance activities to detect ESBL-EB in seafood are needed; salmon and shrimp could be used in initial surveillance activities, representing domestic and imported products.

## 1. Introduction

Antimicrobial resistance (AMR) has been described as a major threat to global health, food security and development [1]. It has far-reaching repercussions, threatening the effective treatment of disease in both human and animal populations and increasing the financial burden on health systems, patients and agriculture industries. It has been estimated that in 2018, AMR cost the Canadian national health-care system 1.4 billion dollars (CAN), which will increase to 7.6 billion by 2050 due to longer hospital stays and prolonged therapies [2]. In 2018, more than 5400 deaths were attributable to AMR in Canada [2]. 

The extended-spectrum β-lactamase (ESBL) producing-Enterobacterales (ESBL-EB) are a group of organisms which demonstrate resistance to third-generation cephalosporins (3GCs) and encompass several important human pathogens such as *Escherichia coli*, *Salmonella* spp. and *Klebsiella pneumoniae*. Third-generation cephalosporins are considered critically important antimicrobials by the World Health Organization (WHO) and of very high importance to human medicine by Health Canada, and are used to treat severe infections where alternatives are limited [3,4,5].

The presence of ESBL-EB in food-producing animals, including aquaculture, has been documented worldwide [6,7,8]. Further, the transmission of ESBL-EB throughout the food chain has been demonstrated in several studies [9,10,11]. In a recent scoping review by Young et al. [12], ESBL-*Enterobacteriaceae* were identified in shrimp and salmon retail products. Additionally, a study of Canadian retail seafood noted the presence of ESBL-EB in 4.2% (52/1231) of various seafood types sampled in Canada, where 67% (35/52) of the positive samples were imported shrimp (N. Janecko, personal communication, 2022).

Shrimp and salmon are the most important seafood products consumed in Canada, being primarily imported and domestic sources, respectively. This is of particular interest in the context of the increasing worldwide exchange of aquaculture products, as antimicrobial use (AMU) varies widely depending upon the country-of-origin. With AMU being a principal driver for AMR, differences in authorized molecules and diverse regulatory oversight are important considerations for AMR in shrimp and salmon. The importance of this food safety issue in Canadian retail seafood has yet to be evaluated within the framework of a formal risk analysis process.

This risk profile used methods described in the Guidelines for Risk Analysis of Foodborne Antimicrobial Resistance (herein denoted the “Codex Guidelines”) adopted by the Codex Alimentarius Commission [13], which is similar to publications by Carson et al. [14] and Loest et al. [15], as a first step in the assessment of the risk of ESBL-EB in retail salmon/shrimp to human health, i.e., a risk profile was undertaken.

Whole genome sequencing (WGS) information was given special consideration in the section concerning risk management. The discernment of AMR movement in the microbiome/resistome, is an important component to the risk profile/risk analysis process. The relationships between AMR genotypes, plasmid profiles, bacterial phylogeny and bacterial genomics contribute knowledge necessary for understanding the interconnection between humans, animals and the environment (i.e., One Health). 

The presence of ESBLs in Enterobacterales in salmon and shrimp available for purchase by consumers in Canada was the AMR food safety issue explored in this risk profile and comprised two principal objectives. The first was to describe the current state of knowledge of ESBL-EB in relation to Canadian retail shrimp and salmon, including factors that may influence the potential risk posed, current and proposed risk management strategies and the relevant policy context. The second was to identify subsequent risk analysis steps to provide direction for policy makers concerning risk management decisions, the necessity of further qualitative or quantitative risk assessments and identifying the need for additional data gathering if necessary for this AMR food safety issue.

## 2. Materials and Methods

The AMR food safety issue explored in this risk profile is the presence of ESBLs in Enterobacterales in salmon and shrimp available for purchase by consumers in Canada. Unlike the other two Canadian AMR risk profiles previously published using the Codex Guidelines [14,15], this risk profile focused on a resistance pattern expressed by one or more genes for multiple bacterial species. The Codex Guidelines, in particular its Appendix 1, was used as a template for the development of this risk profile. The suggested elements for inclusion were utilized and combined for clarity and brevity if applicable [13]. The methodology was similar to that used by Loest et al. [15]. Several data sources were exploited in the preparation of this manuscript including the Canadian Integrated Program for Antimicrobial Resistance Surveillance (CIPARS), Fisheries and Oceans Canada, Statistics Canada, Agriculture and Agri-Food Canada, peer-reviewed literature, grey literature and expert opinion. Peer-reviewed literature and grey literature were identified by performing an exhaustive search via the internet using multiple web browsers and publicly available databases including PubMed, PubMed Central, Web of Science, Scopus and Google Scholar. Various key words were selected on the basis of an expert opinion and the need of this specific risk profile, such as Enterobacterales, *Enterbacteriaceae* and the related individual species, seafood, salmon, shrimp, aquaculture, ESBL and specific ESBLs, antimicrobials, specific drugs and/or resistance. Unless indicated otherwise, the information was specific to the Canadian situation. Non-Canadian data were used where pertinent and informative (e.g., when there were identified gaps in Canadian data) [15]. 

As described by Loest et al. [15], the data for each major section of the risk profile were assessed in terms of data quality where appropriate as follows: (1) applicability of the data within a Canadian context based on the geographic location of information collection; (2) type of information (e.g., surveillance data, peer reviewed publications, reviews); and (3) year of data collection or publication. Scores across the subsections of information were summed up and then averaged to provide an overall measure of data quality per section. As per previous Canadian AMR risk profiles [14,15], to help advise policy makers, each major section of the risk profile was also categorized into “levels of concern” (1, 2 or 3), with 1 being the lowest level of concern and 3 the highest. These take into consideration the interpretation and meaning of the findings for each section, where appropriate (e.g., medical significance of the antimicrobial in question).

Details for the data quality evaluations are provided in Appendix A in Appendix A and Excel file S1.

## 3. Results (Headings as per the Codex Guidelines)

### 3.1. Description of the AMR Food Safety Issue

The AMR food safety issue [13] examined in this risk profile is the presence of ESBLs in Enterobacterales in salmon and shrimp available for purchase by consumers in Canada. 

The Enterobacterales, and more specifically, the ESBL-EB, were selected as this bacterial family encompasses several important human pathogens. They have been reported in most terrestrial and aquatic animal production systems and are commonly found at all levels of the seafood-to-fork continuum [16,17]. In an analysis of internationally reported foodborne outbreaks between 1988 and 2007, members of the Enterobacterales were associated with 30% of 277 outbreaks due to seafood [18]. In the USA, a recent review observed that between 1998 and 2018, among 709 fish associated disease outbreaks with a confirmed etiology; 15 (2.1%), three (0.4%) and one (0.1%) were attributed to *Salmonella*, *Shigella* and *E. coli*, respectively. Three of them involved salmon, two due to *Salmonella* in smoked fish and one due to *E. coli* O157:H7, resulting in 19 and seven illnesses, respectively [19,20]. Finally, again in the USA, a *Salmonella* Weltevreden outbreak affecting nine people in four states was traced to frozen cooked shrimp which resulted in three hospitalizations and a product recall [21].

Salmon and shrimp were the food products examined here, both predominantly produced in an aquaculture environment. They are the most commonly consumed seafood in Canada and may represent a potential exposure source to foodborne bacteria for people consuming seafood. Atlantic salmon (*Salmo salar*) is the principal species of salmon consumed by Canadians [22]. It is primarily produced domestically and its production is subject to strict regulations (e.g., site selection, quality control and approved AMU). The major cultured shrimp species is the whiteleg shrimp (Pacific white shrimp; *Litopenaeus vannamei*) which is a product grown and imported from Central America and Southeastern Asia [23]. The aquaculture regulatory structure and enforcement may differ among importing and exporting countries, including the antimicrobial classes authorized for use. For example, in a survey of four Asian countries, ten different classes of antibiotics were reported being used in finfish, shrimp or prawn production, in contrast to three classes approved for Canadian aquaculture [24,25]. This is an important element to consider when examining AMR development in the seafood-to-fork continuum.

The antimicrobials under scrutiny for this risk profile, are the 3GCs, which are classified by the WHO as ‘Highest Priority Critically Important Antimicrobials’ and by Health Canada as ‘Category I-Antimicrobials of Very High Importance’ [4,26]. The 3GCs are used to treat serious infections caused by multidrug-resistant Enterobacterales for which few treatment alternatives exist, and such infections may result from the transmission of Enterobacterales, including *E. coli*, from non-human sources [26,27]. Resistance to 3GCs can be mediated by ESBL genes. ESBL-EB can exhibit high levels of cross-resistance among cephalosporin formulations, as they demonstrate high-level β-lactamase activity against both human (e.g., cefotaxime, ceftazidime and ceftriaxone) and veterinary (e.g., ceftiofur) antimicrobial active ingredients. Enterobacterales displaying resistance to 3GCs have been isolated from seafood products and the aquaculture environment in Canada and other countries [12,28,29,30,31] (N. Janecko, personal communication, 2022). Their ubiquity and inter/intra species genetic promiscuity make them important as an antimicrobial resistance gene (ARG) reservoir and a key player in the circulation of resistance genes.

### 3.2. Information on the AMR Microorganism(s) and/or Determinant(s)

#### 3.2.1. Characteristics of the Identified Foodborne Microorganism(s)

##### Sources and Transmission Routes

The consumption of contaminated retail salmon and shrimp is the principal route of transmission of ESBL-EB considered in this risk profile. Studies examining AMR, bacterial flora and contamination in the seafood-to-fork continuum often examine Enterobacterales, even though most are not considered commensals or pathogens of cultured aquatic animals [15]. Enterobacterales’ seemingly ubiquitous presence in harvested aquatic animals is believed to be due to the terrestrial contamination of the aquatic environment and/or contamination at harvesting, processing and retail [12,32,33,34,35] (N. Janecko, personal communication, 2022). 

##### Pathogenicity, Virulence and Linkage to Resistance of Particular Strains

ESBL producers of seafood origin are mostly reported as *E. coli*, which is part of the normal diversified microbiota of the gastrointestinal tract of humans and animals. However, there is a diversity of *E. coli* strains that possess pathogenicity and virulence, causing intestinal and extra-intestinal diseases, including life-threatening complications [36,37]. A range of mechanisms, such as the production of adhesins and toxins, contribute to the *E. coli* pathogenicity features, which affect bacterial colonization and fitness. The virulence factors are produced by host bacteria via chromosomal genes and plasmids [37]. ESBL *E. coli* producers belong to the seven major phylogenetic groups, A, B1, B2, C, D, E and F, and thus there may be little correlation between the phylogroup distribution and ESBL production [38,39,40,41].

Virulence factors are yet to be identified on ESBL-bearing plasmids, but have been observed in various ESBL producers from seafood and aquatic sources. In one study, multidrug-resistant CTX-M-1-producing *E. coli* isolates of seafood-origin were typed phylogenetically to be phylogroups A and B1, and carried virulence genes *fimA* (encoding type 1 fimbriae), *aer* (for aerobactin iron uptake system) and *papGIII* (adhesin PapG) [42]. Up to 13 virulence genes were detected in *E. coli* belonging to one of 14 different clones producing CTX-M-9, CTX-M-15 or TEM-24; for example, these genes were *afa* (adhesin), *fimH* (mannose-specific adhesin of type 1 fimbriae), *focG* (F1C fimbriae), *iutA* (aerobactin siderophore receptor), *irp2* (yersiniabactin), *kpsM* KI (capsule synthesis), *papA*, *papC*, *papE* and *papG2* (encoding P fimbriae), *sfaS* (S fimbriae) and/or *traT* (serum resistance-associated factor) [39]. ESBL-producing colistin-resistant *E. coli* of fish and shrimp-origin were found to possess the virulence gene *astA* which encodes the enteroaggregative *E. coli* heat-stable enterotoxin and is therefore indicative of diarrheagenic *E. coli* [43,44]. Another study also revealed multidrug-resistant ESBL-producing *E. coli* from a fish source to be positive for several virulence genes; specifically: *gad* (for glutamate decarboxylase), *iss* (for increased serum survival), *lpfA* (long polar fimbriae), *nfaE* (diffuse adherence fimbrillar adhesin) and/or *vat* (vacuolating autotransporter toxin) that may be dependent on the phylogroups of the isolates [45]. A multidrug-resistant SHV-12 ESBL- and VIM-1 carbapenemase-producing *E. coli* isolate (sequence type ST10) from retail seafood was found to carry the *gad* gene [46]. The latter is part of core virulence genes commonly observed in ST10 from human and food animal sources [47]. Multidrug-resistant CTX-M-15-producing *K. pneumoniae* of retail fish belonged to a hypervirulent capsular serotype with the *wzi* gene encoding an outer membrane protein involved in capsule attachment to the cell surface. The virulence gene *iutA* was prominent in all isolates while the yersiniabactin-encoding gene *ybtS* and ferric uptake operon associated gene *kfu* were confirmed in two *E. coli* isolates [48]. Additionally, the presence of ESBL plasmids in some pandemic sequence types of *E. coli* was found to have enhanced the competition fitness and serum resistance of the *E. coli* host [49], suggesting that the carriage of ESBL itself can be considered a virulence factor. 

##### Growth, Survivability and Inactivation in Foods (e.g., D-Value, Minimum pH for Growth) of Foodborne AMR Microorganisms in the Food Commodity Production to Consumption Continuum

As noted by Loest et al. [15], seafood contamination by Enterobacterales can occur at multiple points along the seafood to fork continuum, from the culture environment, harvesting and processing activities and seafood preparation in the retail and consumer home environment [15,50,51]. 

Although physicochemical parameters of seafood have not been reported to affect AMR, they can impact bacterial prevalence [15]. Limits for different physicochemical parameters of seafood have been outlined by the US Food and Drug Administration (FDA) for which growth of the bacterial pathogens *E. coli*, *Salmonella* and *Shigella*, can occur [52]. These species demonstrate similar tolerance ranges or limits for selected variables including temperature, 5.2–48 °C; pH 3.7–10.0 and water activity (Aw) 0.94–0.96, which may vary from non-pathogenic bacterial strains. Supplemental Appendix A provides additional information.

Seafood exposure to ambient temperatures can increase bacterial growth, however, even at lower temperature ranges, bacterial growth may occur. As also noted in Loest et al. [15], in a study examining the microbiology quality of fish, the quantity of *E. coli* in salmon increased after two days of storage at 4 or 8 °C, with no differences reported between the two temperatures [15,53]. Additionally, temperature exposure may affect bacterial pathogenicity. A study of *Salmonella* growth in seafood at varying temperatures demonstrated not only increased growth, but also an increased expression of virulence genes in non-typhoidal and typhoidal *Salmonella* isolated from fish held at room temperature, in comparison to seafood held at 4 °C [54].

Thermal resistance of bacteria in food depends upon the salt and A_w_, type of food product or matrix and the bacterial strain in question [55]. The D-10 values (the time required to reduce the microbial population by 90% at a specific temperature) reported for Enterobacterales in seafood are similar to what is found in other food products [56]. A relationship between the mild thermal treatment of shrimp products to control bacterial growth and the selection of resistant bacterial strains was proposed in a USA study examining AMR in wild-caught and farmed retail shrimp [57].

For salmon, wild, hybrid and farmed, pH values for fish flesh occupy a narrow range between 6.42 and of 7.18, whereas for shrimp, the pH values range between 6.42 and 6.8 [58,59,60]. These values indicate that Enterobacterales contamination would not be inhibited at normal pH values encountered in fish and shrimp tissues. 

A_w_ refers to the water present that is not bound to food molecules (i.e., seafood) and available for bacterial growth. Examples of A_w_ values for various salmon products include: Fresh Chilean Raised Salmon Fillets, 0.985; Canned Salmon Paste, 0.970; Smoked Salmon 0.965; and Smoked Salmon Pate, 0.993 [61]. As noted by Loest et al. [15], the reduction of the value of this parameter, i.e., decreasing the amount of available water, is important for prolonging shelf life and ensuring seafood quality, where most bacterial growth can be minimized when A_w_ levels are held below 0.85 [62].

##### Distribution, Frequency and Concentrations of the AMR Hazard(s) in the Food Chain

Enterobacterales and ESBL-EB in food animals and in the food chain have been described for several species including fish and seafood [10,30,63,64]. The presence of *E. coli* specifically in the aquaculture farm, harvesting and processing environments has been previously detailed [15]. 

(i).Farm and harvesting levels

At the farm and harvesting levels, Enterobacterales are found in source waters, the culture environment and in the aquacultured animals themselves. The presence of Enterobacterales is associated with the contamination of the aquaculture environments. Certain species such as *E. coli* and *Salmonella* can survive in the aquatic environment for relatively long periods depending on water temperatures and physicochemical variables [65,66]. Natural water bodies have been shown to be reservoirs of ARGs including ESBLs, which can be an important factor in ARG exchange [67]. Of the two types of seafood considered in this risk profile, shrimp contaminated with Enterobacterales are described in the literature at the farm-level [15], whereas this is not the case for salmon. Higher culture temperatures, contaminated water sources and in some cases, integrated culture conditions associated with shrimp culture have been cited as causes of Enterobacterales presence [68,69,70]. The majority of salmon consumed in Canada is domestically produced. Canadian aquaculture operations may be subject to more stringent regulatory measures concerning AMU and the placement of aquaculture operations may avoid anthropogenic influences or contamination.

Although the more general bacterial groupings of fecal coliform, total coliform and Enterobacterales are often described, several specific species of this family are identified in some studies examining shrimp or salmon, including *Salmonella*, *Enterobacter cloacae*, *K. pneumoniae*, *Citrobacter freundii* and *Enterobacter aerogenes*, among others [64,71,72]. Supplemental Appendix A summarizes information concerning the prevalence of Enterobacterales in shrimp and their culture environment. 

ESBL-EB have not only been identified in the aquatic environment in relation with human activity such as hospital and sewage effluents, for example, but also in natural waters such as rivers, lakes and seas [45,73,74,75,76,77,78,79]. Several publications have demonstrated the presence of ESBL-EB in the aquacultured environment, the animals and the products (Supplemental Appendix A).

In a shrimp-focused study from India, out of 37 shrimp farms, eight were positive for ESBL-*E. coli* and five were positive for ESBL-*K. pneumoniae*. Two CTX-M types were identified in 32 *E. coli* isolates (CTX-M1/CTX-M9) in the sediment (15%/3%), water (3%/0%) and shrimp (8%/8%), whereas in 15 *K. pneumoniae* isolates, only CTX-M 1 was found in the water (5%) and shrimp (13%) [7].

(ii).Processing

Several studies have been published investigating microbial contamination in shrimp processing plants, whereas none were identified concerning salmon. Due to the number of different products, direct comparisons between studies are difficult. The products have included raw (fresh or frozen), peeled, peeled deveined, headless and block frozen or individually quick-frozen shrimp, among others. There are varying degrees of widespread contamination of Enterobacterales at this level of production. This has been attributed to a poor quality of raw materials and inattention to personal hygiene during production and handling [80,81,82]. Supplemental Appendix A summarizes studies examining this aspect of contamination of the seafood to fork continuum with Enterobacterales.

(iii).Retail

The last link in the seafood chain examined is the retail level. Here, we find that a seafood microbiota is the culmination of the previous production and the processing activities and control measures. Loest et al. [15] conducted a detailed review of *E. coli* in retail salmon and shrimp, as well as the presence of varied AMR phenotype/genotypes. *Enterobacteriaceae*, coliforms and *Salmonella* are the most frequent Enterobacterales reported in the literature examining prevalence and bacterial concentration in seafood. A recent US-based publication by Tate et al. [83] looking at the prevalence of various non-Enterobacterales and *Salmonella* identified *Salmonella* in 1/501 and 2/498 salmon and shrimp samples, respectively. These included Newport and Teko serovars [83]. *Salmonella* identified in seafood in foreign retail markets are not among the most common serovars associated with human infections in Canada (Enteritidis, Typhimurium, I 4,[5],12:i:- and Infantis), but most have been recognized as causing human disease in Canada, even if at a lower level [84,85,86,87]. Supplemental Appendix A summarizes information concerning Enterobacterales described in shrimp and salmon at the retail level.

For the presence of ESBL-EB in retail shrimp and salmon, a recent scoping review of the literature by Young et al. [12] examined ESBL-EB in salmon and shrimp. The studies retained in the scoping review were regarding the retail sector, and the majority identified ESBL-EB in shrimp (12/16), whereas only 2/16 found ESBL-EB in salmon. Several bacterial genera were identified as harboring ESBL genes, where *bla*_CTX-M_ gene types were the most commonly reported. *E. coli* and *Klebsiella* spp. were the principal bacterial species, frequently found in shrimp products originating from Asia [12].

Among additional studies identified, in Vietnam, Le et al. [44] found ESBL-producing *E. coli* in 20% (22/112) of raw retail shrimp examined. A study from the USA, assessing AMR in wild caught and farmed retail shrimp, found *bla*_CTX-M_ and *bla*_SHV_ genes (43% and 6%, respectively) in wild caught shrimp, and not in imported farmed shrimp [57]. Finally, a publication in preparation examined ESBL-EB in imported and domestic food products purchased through retail in Canada (N. Janecko, personal communication, 2022). It demonstrated an overall prevalence of ESBL-EB of 4.2% (52/1231) in fish and seafood. Shrimp were the source of over half (63%, 33/52) of all ESBL-EB isolates identified in the study, principally *E. coli, K. pneumoniae* and *Enterobacter* spp. In shrimp and seafood, the *bla*_CTX-M_ genes were the most common ESBL determinants where *bla*_CTX-M-15_, *bla*_CTX-M-27_ and *bla*_CTX-M-55_ accounted for 30, three and 14 ESBL-EB isolates, respectively. In addition, *bla*_SHV-2_ (1/52), *bla*_SHV-12_ (3/52) and *bla*_SHV-28_ (3/52) were identified in a limited number of samples (N. Janecko, personal communication, 2022).

#### 3.2.2. Characteristics of the Resistance Expressed by the AMR Microorganism(s) and/or Determinant(s)

##### Resistance Mechanisms and Location of AMR Determinants

The bacterial species of seafood-origin include foodborne bacteria (e.g., *E. coli*, *Salmonella* spp., *K. pneumoniae* and other Enterobacterales) and aquatic bacteria such as *Vibrio* spp. In these Gram-negative bacteria, the production of ESBLs is the predominant mechanism for high-level resistance to β-lactams, including extended-spectrum β-lactams (especially the oxyimino-cephalosporins). ESBLs are mostly encoded on plasmids and are, less frequently, present in the chromosome [88]. The ESBL plasmids from seafood-derived bacteria belong to a number of incompatibility groups, including IncA/C, IncA/C2, IncF, IncHI1, IncHI2, IncI1, IncN, IncQ, IncX and IncY [45,46,89,90], which may have a broad host range among multiple Gram-negative bacterial species.

ESBLs usually consist of class A β-lactamases (e.g., TEM/SHV derivatives and CTX-M) and certain class D OXA β-lactamases with CTX-M enzymes most frequently detected in bacteria of human, animal and environmental sources [88,91,92]. The CTX-M enzymes can be mostly clustered into five groups based on sequence homologies: CTX-M-1 (including CTX-M-15 and CTX-M-55), CTX-M-2, CTX-M-8, CTX-M-9 and CTX-M-25 [91]. ESBLs of seafood-origin reported to date comprise CTX-M-1, CTX-M-8, CTX-M-9, CTX-M-14, CTX-M-15, CTX-M-24, CTX-M-27, CTX-M-55, CTX-M-65, CTX-M-130, SHV-5, SHV-12, SHV-32, SHV-186, SHV-187, TEM-24, TME-52 and OXA in *E. coli*, *Salmonella* or other Enterobacterales [38,39,42,44,45,46,64,76,89,93,94,95,96,97,98,99,100,101,102,103]. Additionally, class A PER-1 was identified from *Vibrio parahaemolyticus* of fish. Of these reported ESBLs, CTX-M-15 and CTX-M-55 are dominant [38,39,78,95,97,98,100]. For example, in studies with fish samples of Cambodia, 23 out of 32 (72%) CTX-M-producing *E. coli* isolates produced CTX-M-15 while nine out 10 (90%) CTX-M *Salmonella* spp. expressed CTX-M-55 [97,98].

ESBL producers carrying other non-ESBL β-lactamase genes, such as those encoding AmpC enzymes, are frequently present, which may expand β-lactam cross-resistance phenotypes [104]. The co-presence on plasmids of multiple *bla* genes encoding CTX-M, SHV, CMY-2 and/or TEM-1 was reported in *Salmonella* isolates derived from retail fish [89]. Three *bla* genes encoding SHV-12, AmpC ACC-1 and carbapenemase VIM-1 were co-located in a 194-kb multidrug resistance IncY plasmid of *E. coli* from retail seafood [46]. 

Importantly, ESBL producers of seafood-origin often display multidrug resistance phenotypes because of the co-presence of ESBL genes and other resistance determinants either in the same plasmids or in plasmids and chromosomes [42,90,100], which is implicated in co-resistance selection. In the work by Janecko examining Canadian retail seafood, among the 52/1231 ESBL-EB isolates identified, all were found to be resistant to ≥3 classes of antimicrobials (N. Janecko, personal communication, 2022).

##### Cross-Resistance and/or Co-Resistance to Other Antimicrobial Agents

ESBLs are able to hydrolyze a wide range of β-lactams, including numerous penicillins, monobactams and cephalosporins, resulting in cross-resistance to multiple β-lactams. Yet, a major feature for ESBL-producers is to possess resistance to oxyimino-aminothiazolyl cephalosporins, such as cefuroxime, cefotaxime, ceftriaxone, ceftazidime, cefepime and cefpirome [88,105]. 

Plasmids or other mobile genetic elements encoding ESBLs often contain multiple resistance genes that also simultaneously confer co-resistance to a variety of other structurally-unrelated antimicrobials, including aminoglycosides, amphenicols, polymyxins, quinolones, sulfonamides and tetracyclines [44,48,90,104,106,107]. For instance, a conjugative mega IncHI2 plasmid of 243 kb of *E. coli* from retail food (including shrimp) was found to carry resistance genes *bla*_CTX-M-65_, *mcr-1* (for resistance to polymyxins), *fosA* (fosfomycin), *cmlA* and *floR* (amphenicols), *aadA1*, *aadA2* and *aphA1* (aminoglycosides), *dfrA12* (trimethoprim) and *sul* (sulfonamides) [90]. A retail fish-derived *E. coli* isolate containing *bla*_CTX-M-1_ and *sul2* was found to also carry a class one integron with a gene array of *dfr32-ereA-aadA1* which mediates resistance, respectively, to trimethoprim, erythromycin and streptomycin/spectinomycin [42]. In another study with retail fish, CTX-M-producing *E. coli* isolates were revealed to carry three to five other resistance genes (*tetA* [conferring tetracycline resistance], *dfrA1*, *sul1* and *sul2* [sulfonamide resistance], and *qnrB*, *qnrS* and *aac(6’)-Ib-cr* [quinolone resistance]) and *K. pneumoniae* isolates were highly diverse with a substantial acquired resistance gene profile (*bla*_TEM_, *bla*_SHV_, *bla_O_*_XA-1-like_, *tetA*, *strA*, *strB, dfrA1*, *sul1*, *sul2*, *qnrB*, *qnrS* and/or *aac(6’)-Ib-cr*) [48]. A portion of ESBL producing *E. coli* from fish and shrimp also carried mobile colistin resistance genes (*mcr-1* and/or *mcr-3*), thus possessing the polymyxin resistance phenotype [44,76,90]. Many of these ESBL-producers were also revealed to possess *sul1*, *sul2* or *sul3* genes [76]. A multidrug resistance plasmid of *V. parahaemolyticus* from raw shrimp was found to encode *bla*_PER-1_ and also carried other resistance genes, specifically, the quinolone resistance gene *qnrVC6,* a four-gene cassette conferring resistance to aminoglycosides (*aacA3* and *aadA1*), chloramphenicol (*catB2*) and trimethoprim (*dfrA1*) [94]. Additionally, fecal matter from Gilthead seabream was found to contain ESBL genes, *bla*_SHV-12_ and *bla*_TEM-52_, as well as the genes *aadA*, *cmlA*, *sul* and *tet*A [102]. 

The features of cross-resistance and co-resistance facilitate the selection or enrichment of multidrug-resistant bacteria under the selection pressure of only one relevant antimicrobial agent. In Canada, there is only a limited number of non-β-lactam antimicrobial agents authorized for use in aquaculture (e.g., florfenicol, ormetoprim, oxytetracycline and sulfonamides). However, the use of these agents via water or in-feed medication may select or enrich multidrug-resistant bacterial species carrying relevant co-resistance genetic determinants [108,109].

Antimicrobial drug residues such as macrolides, quinolones, sulfonamides and tetracyclines have been detected in the aquatic environment such as freshwater and lake sediments, with a correlation observed between the presence of drug residues and the prevalence of resistance genes [77,110,111,112]. Furthermore, aquaculture systems including integrated fish farms can be contaminated, even in the absence of AMU, by other sources of resistance determinants or resistant bacteria from animal manures, human wastes and soils [45,69,77,79,110,113,114]. A metagenomics study of integrated and monoculture aquaculture farms confirmed the development and dissemination of a wide range of resistance genes (including those encoding CTX-M, SHV and TEM enzymes) and associated mobile genetic elements [70]. 

##### Transferability of Resistance Determinants between Microorganisms

Genes encoding ESBLs are usually located on transferable plasmids or associated with mobile genetic elements such as insertion sequences and transposons [88,115]. There is no exception for ESBL-producing bacteria of seafood-origin [108]. Thus, the horizontal gene transfer enabled by mobile genetic elements contributes significantly to the spread of resistance determinants among terrestrial and aquatic microorganisms [42,88,94]. In fact, aquaculture systems including coastal waters, lakes, rivers and farms are reservoirs of diverse ARGs and mobile genetic elements, resulting in hotspots for resistance dissemination [70,108,116]. 

ESBL-producing bacteria of seafood origin are known to contain a range of plasmid types (e.g., IncA/C, IncF) and other mobile genetic elements [108]. Plasmid types of seafood-origin, containing integrons and/or transferable or conjugative features, are often shared by bacteria of human and terrestrial animal sources [108,117]. For example, IncF plasmids carrying genes for several CTX-M enzymes were widely found in Enterobacterales from multiple seafood products and were also observed worldwide in other foodstuff and animal species, suggesting the widespread transmission of these plasmids [95]. Another recent study with retail fish revealed that most ESBL-encoded plasmids were transferrable and belonged to IncI1, IncA/C, IncHI1 and IncN [89]. Similarly, several CTX-M-1 plasmids from marine fish were also conjugative with the presence of multiple resistance genes [107]. A CTX-M-1-producing *E. coli* was found to contain the insertion sequence IS*Ecp1* which is closely located to *bla*_CTX-M_ and deemed important for the inter-plasmidal transfer of the ARG as well as movement between plasmids and the bacterial chromosome [42]. Carrying an insertional sequence dubbed “IS*CR1*”, a PER-1 encoding plasmid has been conjugated from its host *Vibro* to a recipient *E. coli* [94]. Overall, the readily transferable capability is a key feature of ESBL gene-bearing plasmids.

#### 3.2.3. Summary of Data Quality and Level of Concern

This section included information regarding the microorganisms, AMR and/or resistance determinants of this AMR food safety issue. The data quality score was 6.0. Most publications consulted were recent and peer reviewed, and some pure science research subjects were considered applicable to the Canadian context (e.g., molecular studies). However, the majority of the manuscripts were regarding research conducted outside of Canada, and the applicability to the Canadian context cannot be assumed, resulting in a reduced score. The level of concern here was evaluated to be a three for the following reasons. The importance of the 3GCs in both human and animal medicine and AMR to these agents are overarching themes. This section speaks to the presence of Enterobacterales and ESBL-EB in multiple aquatic environments, multiple host species and production levels. The implication of resistance determinants present on mobile genetic elements is an important consideration concerning transmission between bacteria and the different aquaculture production compartments, as well as the co-carriage with other ARGs and cross-resistance. Finally, the importation of seafood, primarily shrimp, from countries with divergent AMU and regulatory structures, can have a major impact on AMR or determinant prevalence in seafood and could present a greater likelihood of exposure than domestic products. 

### 3.3. Information on the Antimicrobial Agent(s) to Which Resistance Is Expressed

#### 3.3.1. Class of the Antimicrobial Agent(s)

The cephalosporins are antimicrobials of the β-lactam class which are bactericidal via the inhibition of cell wall synthesis [118]. The first-generation of cephalosporin antimicrobials were first marketed in 1964 with a principal activity against Gram-positive bacteria. Successive generations have followed, characterized by an increased spectrum of activity against important Gram-negative bacterial pathogens. Bacteria with an ESBL phenotype exhibit non-susceptibility to penicillins, first- and second-generation cephalosporins, 3GCs, of which ceftriaxone and ceftiofur are examples used in human and veterinary medicine, respectively, and monobactams. Additionally, the ESBL enzymes are generally inhibited by clavulanic acid [91,119].

#### 3.3.2. Non-Human Uses of the Antimicrobial Agent(s) (Use in Aquaculture)

The use of multiple classes of antimicrobials is well documented in cultured aquatic animals [24,120,121,122,123,124,125,126]. Cephalosporins are not listed and are not recognized as important veterinary antimicrobials for aquaculture by the World Organisation for Animal Health (WOAH), although their use has been reported [127]. As noted by Young et al. [12]), this includes the first-generation cephalosporins, cephalexin and cefradine, used in Vietnam and China and the detection of cefotaxime, and a third-generation cephalosporin in fish ponds in China [12,24,120,128,129]. The sales of antimicrobials intended for use in Canadian aquaculture are quantified by the Veterinary Antimicrobial Sales Reporting (VASR) system, as part of Health Canada and CIPARS, as well as use, quantified by the Fisheries and Oceans Canada National Aquaculture Public Reporting Datasets [130,131]. The legislation in place requires the submission of sales and use data to these two federal surveillance activities, respectively. No β-lactam products are currently labeled for use in Canadian aquaculture, and there has been no evidence of β-lactam sales or use in Canadian aquaculture production [25,130,131,132].

#### 3.3.3. Human Uses of the Antimicrobial Agent(s)

##### Spectrum of Activity, Indications for Treatment and Importance of Antimicrobial Agents including Consideration of Critically Important Antimicrobial Lists

Cephalosporins belong to the β-lactam class of bactericidal antimicrobials and are categorized into five generations based on when they were discovered and their antimicrobial properties. In general, the cephalosporins are effective against Gram-positive bacteria, and the spectrum of activity against Gram-negative bacteria increases with successive generations [133]. The fourth-generation cephalosporins demonstrate activity against *Enterobacteriaceae* and other Gram-negative bacilli-producing AmpC β-lactamases, Gram-positive cocci, Pseudomonads and certain ESBL genotypes [134]. The fifth-generation cephalosporins have a similar spectrum, but are used principally in the treatment of methicillin-resistant *Staphylococcus aureus* and Gram-negative bacteria, other than those producing ESBLs. For the purposes of this manuscript, the activity of ESBLs and resistance expressed to 3GCs will be considered. The 3GCs are extremely important antimicrobials utilized as front-line treatments in a variety of severe infections and are the treatment of choice when the susceptibility of the infectious agent is unknown. Further, they are listed on the WHO essential medicines list, which recommends that their use should be limited to treat specific conditions [135]. The risk profile published by Carson et al. [14] provided a critical description of human antimicrobial use concerning 3GCs, and the reader is referred to this publication for additional details. 

##### Distribution, Cost and Availability

The distribution and cost of the 3GCs can have a direct impact on antimicrobial exposure in the human population and provide a measure of perception of the relative access and availability of these antimicrobials for use in people in Canada. These parameters have remained relatively stable over the last few years. In Canada, public funding of antimicrobials is regulated at the provincial level. Details concerning funding for in-hospital, in-patient and pharmacy dispensing, as well as costs of oral and parenteral formats of the 3GCs, were relatively unchanged in 2020 compared to that reported for 2018 [14]. Cost (in CAD) per unit of cefixime in liquid and tablet formats was $0.39/mL and $2.72–$3.39/tablet, respectively [136,137,138,139]. Parental ceftriaxone and cefepime costs identified were $12.49/g active ingredient and $15.10/g active ingredient, respectively [136,139]. 

##### Availability of Alternative Antimicrobial Agents

Carbapenems, namely imipenem and meropenem, are considered the antimicrobials of choice against ESBL-producing bacteria and are often used to treat serious ESBL-producing Enterobacteriaceae infections in people [140,141]. However, with increased carbapenem use, carbapenem-resistant Enterobacterales have emerged and interest in effective alternatives to manage these infections has subsequently developed [142].

ESBLs are usually inhibited by β-lactamase inhibitors, such as clavulanic acid. Therefore, β-lactam/β-lactamase inhibitor combinations such as amoxicillin–clavulanic acid can be used to treat infections due to ESBL-producing organisms. The use of newer β-lactam/β-lactamase inhibitor combinations such as ceftolozane–tazobactam, which has been approved for use in Canada, can be effective carbapenem sparing therapies [143,144,145]. 

##### Trends in the Use of Antimicrobial Agent(s) in Humans

The IQVIA institute compiles information concerning human antimicrobial consumption in Canada, providing insight into community pharmacy dispensing and hospital purchasing. The Canadian Antimicrobial Resistance Surveillance System (CARSS) analyzes the IQVIA data and reported an overall increase in the quantities of antimicrobials intended for use in people between 2014–2018, including community and hospital settings, of 17.3 to 17.5 Defined Daily Doses (DDDs) per 1000 population days [146]. In 2018, the largest proportion of AMU took place in the community, where 90% of the DDDs were dispensed compared to 10% for hospitals; however, the 3GCs were not among the top 10 most commonly prescribed antimicrobials [146]. Conversely, ceftriaxone was the fifth most purchased antimicrobial across all Canadian hospitals from 2010 to 2016, and ceftriaxone consumption was shown to increase from 0.06 to 0.11 DDDs/1000 inhabitant-days [147]. Although ceftriaxone was not among the top five most purchased antimicrobials in 2018, the amount of ceftriaxone purchased by Canadian hospitals remained relatively similar at 50.6 DDDs annually/1000 inhabitants [146]. It should be noted that trends in antimicrobial purchases by hospitals do not necessarily reflect the trends in use, as factors related to cost and use may differ [14]. 

Based on the Canadian Nosocomial Infection Surveillance Program (CNISP), AMU surveillance data from sentinel hospitals from 2009–2016 revealed increasing trends in 3GC use. In adult-only hospitals, 3GC (ceftriaxone) use increased by 85% (from 39 DDD/1000 patient-days in 2009 to 55 DDD/1000 patient-days in 2016, *p* = 0.02), although this did not represent a large proportion of the total AMU (9.7%) [148]. The CNISP also reported that in adult intensive care units in 2016, ceftriaxone was among the top five antimicrobials used at 119 DDD/1000 patient-days and the third most used antimicrobial at 49 DDD/1000 patient-days [148]. In another CNISP publication, Liang et al. [149] compared three point prevalence surveys between 2002 and 2017. Among all patients receiving an antimicrobial, an increase was reported for 3GC use between 2002 and 2017, from 12.8% to 17.5% (*p* < 0.0001), respectively. When examining AMU data, among 21 CNISP hospitals during the 2018 surveillance period, ceftriaxone remained in the top five antimicrobials distributed in both non-intensive care units (39.1 DDDs per 1000 patient-days) and intensive care units (133.3 DDDs per 1000 patient-days) [146]. 

Finally, a 2017/18 AMU survey of pediatric inpatients within the CNISP hospital network revealed that 3GCs were the antimicrobial class with the highest use (84 days of therapy/1000 patient-days) and this has remained relatively constant since 2017. This is followed closely by extended-spectrum penicillins (80 days of therapy/1000 patient-days) [150]. 

#### 3.3.4. Summary of Data Quality and Level of Concern

The data quality score for this section is 8.0. There is excellent and recent information regarding AMU in the Canadian aquaculture industry. However, there is a lack of information concerning AMU practices for imported seafood products, which leads to uncertainty concerning risk associated with AMU and AMR development. Canadian information regarding distribution, costs, availability and AMU trends in the human population is also well detailed leading to a higher quality score. Although the availability and costs of 3GCs have remained stable in recent years, an increase of 3GC use has been reported by both CARSS and CNISP surveillance programs in hospitals from 2010–2016 and 2009–2016. This suggests an increased reliance upon these essential medicines and thus a higher AMU exposure in the human population. These factors lead us to attribute a level of concern of three.

### 3.4. Information on Food Commodity(ies)

#### 3.4.1. Sources (Domestic and Imported), Production Volume, Distribution and per Capita Consumption of Foods or Raw Material Identified with the AMR Hazard(s) of Concern

Canadian seafood production is composed of commercial sea fishery landings and aquacultured products, which totaled 890,054 metric tons (MT) valued at 3.6 million CAN in 2020 [151]. Shellfish, groundfish and pelagic species derived from commercial fisheries are responsible for approximately 80% of this value, with shrimp and salmon captures contributing 68,500 MT/$263 million and 8406 MT/19 million CAN, respectively [152,153]. Canadian aquaculture produces 170,805 MT of seafood valued at approximately $1 billion CAN annually. Salmonid aquaculture on the West and East coasts is the dominant production type, responsible for 70% of all aquaculture production, followed by shellfish at 18% [153].

Similar to values reported by Loest et al. [15], Canada imported 572,764 MT of fish and seafood products in 2021 with a value of $4.6 billion CAN, an increase of 10% over 2020 [151]. The two most important imported seafood products were salmon (75,304 tons) and shrimp (60,077 tons), which accounted for almost 24% of all seafood imported. As described by Loest et al. [15], shrimp and salmon consumed in Canada are primarily farmed products, where Pacific white shrimp and Atlantic salmon are the principal farmed species accounting for approximately 76% and 90% of the worldwide production, respectively [23]. In 2020, most shrimp were imported as frozen products whereas salmon imports consisted primarily of frozen or fresh filets [154]. The three most important seafood suppliers to Canada in 2020 were the USA, China and Vietnam, representing 52% of Canada’s total seafood importations [154]. Vietnam (36%), India (26%), China (11%) and Thailand (9.5%) were the most important suppliers of shrimp, while farmed salmon was imported principally from the USA (54%) and Chile (27%). A seeming paradox to note is that Canada exported 90,151 MT of Atlantic salmon and imported 75,304 MT in 2021. This can be explained by higher profit margins from salmon exportation and the lower costs of salmon imports. 

Domestic cold water shrimp fishery harvests are essentially destined for exportation, whereas imported warm water shrimp are the principal products eaten in Canadian households [155]. Canadian household consumption of fish and seafood rose by 5.6% between 2017 and 2021 [156]. Canadians spend the majority of their seafood budget on salmon ($45 CAN/year) and shrimp ($45 CAN/year) and in 2015, the percentage of the Canadian population that consumed salmon and shrimp daily was 5.2% (92.42 g/day) and 3.7% (36.38 g/day), respectively [155,157]. 

#### 3.4.2. Characteristics of the Food Product(s) That May Impact Risk Management (e.g., Further Processed, Consumed Cooked, pH and Water Activity)

Growth, survivability and inactivation of bacteria in seafood (e.g., D-value, minimum pH for growth) were examined previously. As mentioned, normal physicochemical parameters (e.g., pH, A_w_) of seafood are not inhibitory to Enterobacterales and storage temperatures between 6.5 °C and 49.4 °C can contribute to bacterial growth. This exemplifies the importance of cold-chain maintenance and hygiene and the application of HAACP programs in seafood manipulation [158]. 

The source of seafood can affect pre-harvest contamination with bacteria and/or important ARGs, and may be associated with an increased risk, as addressed previously. Enterobacterales/ARG contamination of seafood arriving from such an environment may persist throughout processing. 

Seafood is sold in many forms including fresh, frozen, cooked, smoked and canned, among others [159]. Although cooked seafood poses little health risk, cross-contamination at the retail and consumer level is possible where raw seafood is utilized. Seafood consumed raw, such as sushi, has been associated with bacterial foodborne outbreaks and requires extra care in preparation [21,160].

The seafood type or form may also be of importance. The manipulation of seafood during processing can have a noticeable impact on bacterial contamination. For example, Bandekar et al. [161] compared prawn products and found that 42% of the headless prawn were contaminated with *Salmonella,* whereas no *Salmonella* were found in the whole prawn. Moretro et al. [162] demonstrated that processing fish filets manually resulted in lower contamination with spoilage bacteria when compared with industrialized methods. 

The bacterial species present, even at low levels, can be of concern. The Bacteriological Guidelines for Fish and Fish Products administered by the Canadian Food Inspection Agency define maximal *E. coli* levels permitted in ready-to-eat and fresh seafood [163]. Certain strains of bacteria such as *Salmonella* or *Shigella* are infective at relatively low doses and can represent a health risk [158,163]. 

#### 3.4.3. Description of the Food Production to Consumption Continuum (e.g., Primary Production, Processing, Storage, Handling, Distribution and Consumption) and the Risk Factors That Affect the Microbiological Safety of the Food Product of Concern

The recent risk profile by Loest et al. [15] contained a detailed description of the food production-to-consumption continuum for these host species and associated risk factors and readers are referred to this publication for a more detailed discussion. The major risk factors described which affect the microbiological safety of seafood include the elements which promote the development, selection and mobilization of ARGs. These factors identified in the aquaculture production context encompass; culture methods, biosecurity, AMU and treatment methods; the accumulation of antimicrobial residues in the environment; and ARGs from terrestrial sources including agricultural and human activity. Further, at the processing, retail and consumer levels, the elements of cross-contamination, seafood preservation and hygiene were also addressed.

#### 3.4.4. Summary of Data Quality of Level of Concern

The quality score for this section is 6.9. This reflects the information available for seafood production and sources, as well as characteristics of the food products which may impact risk management. The description of the food production continuum was not included in this evaluation, and the reader is referred to a previous risk profile [15] that details this section. The data available for domestic and imported shrimp and salmon were of good quality. Salmon and shrimp are among the most commonly consumed seafood products in Canada, however, more granular information concerning distribution and consumption practices (e.g., cooked vs. raw) were not available. Similar to the risk profile by Loest et al. [15], the lack of knowledge of production practices in countries exporting seafood to Canada remains. Additionally, cold chain maintenance of imported seafood and bacterial contamination/cross-contamination associated with the different stages of harvesting/processing/retail and consumer manipulation need to be considered. The level of concern is estimated at 2.5 due to the increasing consumption of seafood and uncertainties associated with the seafood production continuum.

### 3.5. Information on Adverse Public Health Effects

Characteristics of the disease caused by the identified foodborne AMR microorganisms or by pathogens that have acquired resistance determinants via food.

#### 3.5.1. Trends, Prevalence and Nature of AMR Foodborne Disease in People 

ESBL-EB cause many types of human illness, including urinary tract infections (UTIs), intra-abdominal infections, pneumonia and gastrointestinal infections [2]. They were first detected within hospital settings, but are now commonly transmitted within the community [164]. Information published in the USA indicates that the infections with ESBL-EB were split evenly between community associated infections (CAI) and hospital associated infections (HAI) and long term care [141]. ESBL production has been most frequently reported in *E. coli* and *K. pneumoniae*, due to the detection of few clinical isolates from other Enterobacterales [165]. Initially, nosocomial *K. pneumoniae* infections of the *bla*_TEM_ and *bla*_SHV_ genotypes were the primary source of ESBLs [166]. More recently, *E. coli* carrying the *bla*_CTX-M_ genes have been associated with increased community transmission, particularly causing UTIs [166,167].

Between 2007 and 2016, the proportion of ESBL-producing *E. coli* and *K. pneumoniae* isolates from Canadian hospital sites increased from 3.9% to 12.8%, and from 1.0% to 13.6%, respectively [168]. For both *E. coli* and *K. pneumoniae*, there was a significant increase in the proportion of ESBL-producing isolates from inpatient samples and outpatient samples, as well as for bloodstream infections [168]. An increasing trend in ESBL-producing isolates was also reported in urinary tract and respiratory infections, but this increase was only statistically significant for *E. coli* [168].

In a recent study of Canadian tertiary care hospitals, the proportion of different ESBL genotypes from 2007 to 2018 in *E. coli* and *K. pneumoniae* isolates was examined using CANWARD surveillance data [169]. Of 9588 *E. coli* and 3056 *K. pneumonia*e isolates, 671 (7%) and 141 (4.6%) were found to demonstrate an ESBL phenotype, respectively. The ESBL gene distribution identified included *bla*_CTX-M_ (614/671, 92%), *bla*_SHV_ (15/671, 2.2%) and *bla*_TEM_ (16/671, 2.4%) for *E. coli*; and *bla*_CTX-M_ (102/141, 72.3%), *bla*_SHV_ (36/141, 25.5%) and *bla*_OXA_ (2/141, 1.4%) for *K. pneumoniae,* with the remainder unknown [169]. 

Within Toronto intensive care units and emergency departments, an increase in ESBL-*E. coli* cases co-occurred with an increased proportion of the ST131 *E. coli* clonal type. ST131 represented 31.6% of ESBL-*E. coli* isolates in 2006 and 71% in 2016 [165]. The ST131 *E. coli* clonal type often carries the *bla*_CTX-M_ genotype, and is known to be a highly virulent clone, commonly causing extra-intestinal infections, including bacteremia and UTIs [170,171]. 

Provincial health laboratories report on trends and prevalence of resistance to 3GCs as an indicator of ESBLs in healthcare and community settings. Between 2007 and 2014, the British Columbia Center for Disease Control reported an increase in the proportion of ESBL-*E. coli* isolates and a decrease in ESBL-producing *K. pneumoniae* and *P. mirabilis* isolates from community-based laboratories [172]. In 2018, Public Health Ontario reported that 9.0% of *E. coli* and 5.5% of *K. pneumoniae* specimens tested were resistant to 3GCs in samples from community and hospital-based laboratories. These values reflected a slight decrease in resistance to 3GCs in *E. coli* and no change in *K. pneumoniae* between 2015 and 2018 [173].

There are also data on ESBL-EB from point prevalence studies. According to a study by CNISP, the point prevalence of ESBL-producing healthcare-associated infections increased from 0.4% in 2002 to 2.8% in 2017. In particular, an increase was identified in the percentage of ESBL-*E. coli* infections causing UTIs, which increased from 0% in 2002 to 3.6% (*p* = 0.01) in 2017 [174]. A study of Canadian acute care hospitals found that 1.4% adult inpatients had ESBL-EB colonization or infection on a single day in 2016, an increase from 1.2% in 2012 [33]. 

The 2019 CDC Report on Antibiotic Resistance Threats in the USA identified ESBL-EB as a serious threat to human health, and estimated 197,400 hospitalizations and 9100 deaths occurred in 2017 in the USA due to ESBL-EB infections [141]. The report noted more complex treatments are required for ESBL-EB infections, including hospitalization and reliance on carbapenems. Similarly, the Public Health Agency of Canada has included ESBL-EB (*Klebsiella* spp., *Enterobacter* spp. and *E. coli*) in the highest priority group for antimicrobial-resistant disease threats in Canada based on the total risk score [175]. 

#### 3.5.2. Epidemiological Pattern (Outbreak, Sporadic), Regional, Seasonal or Ethnic Differences in the Incidence

##### Regional

Resistance to 3GCs was first reported in the early 1980s in Germany and France and then in North America, where isolates harboring variants of the *bla*_TEM_ and *bla*_SHV_ β-lactamase genes demonstrated resistance to cefotaxime [176]. CTX-M β-lactamases have become the predominant ESBL gene type worldwide since the early 2000s. The global dissemination of CTX-M has been aided by the association with the IncF narrow host range plasmids and the human pandemic clones *E. coli* ST131 [91]. Although the *bla*_CTX-M-15_ gene is the most predominant ESBL type identified in human populations, there are some regional differences. For example, *bla*_CTX-M-14_ is more prevalent in some Asian countries and Spain and *bla*_CTX-M-2_ more-so in South America [177]. In animal populations, *bla*_CTX-M-1_ tends to predominate, but the common human ESBL gene types can also be found [178].

In Canada, CANWARD is a national surveillance program present in eight of 10 provinces, wherein 12 tertiary hospitals follow AMR trends in pathogens, including ESBL-EB [169]. In the aforementioned study by Karlowsky et al. [169] examining ESBL-producing clinical isolates of *E. coli* and *K. pneumoniae*, they compared the proportions of ESBL^+^ vs. ESBL^-^ isolates for the different regions of Canada. In the West (Alberta, British Columbia, Manitoba and Saskatchewan), Ontario, Quebec and New Brunswick/Nova Scotia, proportions ranged from 7.1–18.4% and 3.2–13% for ESBL-*E. coli* and ESBL-*K. pneumoniae*, respectively. The highest were noted in Ontario for both bacterial species. A regional variability was demonstrated; however, the repartition of participating hospitals may not be representative of the overall population and could affect the interpretation [179].

##### Seasonal

Seasonal differences of ESBL-EB carriage and excretion in people have also been identified. Wielders et al. [180] have shown that ESBL-EB carriage was highest in the months of August and September, controlling for variables such as travel, age, sex and antibiotic use. This suggests that some factors present in the summer months could influence colonization and also indicates the importance of sampling time frames used for prevalence estimates. A delay of one and three to six months between AMU and the development of AMR has been demonstrated in *E. coli* and/or *K. pneumoniae* infections in American and Dutch studies, respectively [181,182]. Consequently, higher infection rates and an increased AMU in winter months may cause a rise in ESBL^+^ isolates later in the year. A study by Schmiege et al. [183] suggested a more immediate relationship, where variation in ESBL-EB excretion in wastewater treatment plants was found to be higher in the winter months, hypothesized to be caused by increased infections and AMU at this time. Environmental variables such as high temperatures and rainfall (dilution) in warmer seasons may also alter the abundance of ESBL-EB in this microbial community affecting the temporal analysis [184]. 

##### Ethnic Differences 

Ethnicity could influence exposure to, carriage or infection by ESBL-EB. A study of a multi-ethnic community in London, UK examined demographic characteristics of bacteriuric patients with ESBLs [185]. They noted that patients with an Asian ethnic background were at a higher risk of carriage than Caucasian British subjects [185]. ESBL-EB carriage has also been positively associated with international travel in several studies, especially to Indian and Asian or African countries [186,187]. Although travel for pleasure cannot be parsed out from this data, some travel may be due to familial or ethnic ties to the visited regions and have an influence on ESBL carriage. Food consumption habits may also differ among ethnicities, and this may have an impact on ESBL-EB exposure. For example, from 1973 to 1992 almost 22% of the foodborne disease outbreaks in Japan were due to seafood, where fish is an important part of the diet and may be eaten raw. This compares to the USA, Canada and the Netherlands, where only 8% of the foodborne disease outbreaks were due to seafood [158]. 

Seafood consumption rarely contributes to bacterial foodborne disease from the Enterobacterales. However, both *E. coli* and *Salmonella* have been identified in outbreaks in the USA in a publication examining the epidemiology of seafood-associated infections, but were relatively infrequent from 1973–2006 [188]. In a study examining seafood associated diseases in Canada, infections due to *Salmonella,* along with the non-Enterobacterales *Staphylococcus* and *Vibrio*, were also reported [189]. In a recent summary of enteric disease associated with shellfish in Canada, from 1998 to 2019, four of 14 outbreaks of bacterial origin were all attributed to *Vibrio parahaemolyticus* [19]. AMR has been identified in pathogens responsible for seafood outbreaks, but resistance due to ESBLs was not identified [190,191].

#### 3.5.3. Susceptible Populations and Risk Factors

Risk factors for acquiring an ESBL-EB infection include pre-existing health conditions (e.g., diabetes requiring hemodialysis, urinary incontinence, cancer and heart disease), a history of invasive procedures, use of a urinary or central venous catheter, residence in a long-term care facility, prior infection with ESBL, prior antimicrobial therapy, prior hospitalization, increasing age and travel [2,192,193,194,195]. In a Canadian study, females were found to have an increased risk of acquiring community-onset ESBL-*E. coli* infections compared to males, although this is likely due to the increased risk of females experiencing community-onset UTIs compared to males and *E. coli* primarily being isolated from urine [193]. 

Neonates are highly susceptible to mortality following an ESBL-EB infection, with an estimated risk of mortality of 30% or greater [195]. A systematic review and meta-analysis analyzing ESBL-EB colonization in North America estimated that the ESBL-EB colonization rate was 13% for residents in long-term care facilities [196]. A Canadian study estimated that the odds of hospital inpatients developing an ESBL-EB infection were 8.28 times higher for patients with previous admission to a long-term care facility compared to those without prior long-term care facility admission [197].

In Canada, the risk of a hospital inpatient developing an ESBL-*E. coli* or *Klebsiella* infection has also been associated with the length of hospital stay and prior use of first-generation cephalosporins or 3GCs [197]. The odds of an inpatient with prior use of 3GCs developing an ESBL-producing infection were 4.52 times higher than that of a patient without prior use of 3GCs [197]. The odds of an inpatient with prior use of first-generation cephalosporins developing an ESBL-producing infection were 2.38 times higher than that of a patient without prior use of first-generation cephalosporins [197].

In Canada, overseas travel significantly increases the risk of acquiring an ESBL-EB infection by up to 5.7 times, with travel to the Indian subcontinent contributing the largest risk [2,193]. 

#### 3.5.4. Burden of Illness: Impact on Outcomes

Numerous studies examining ESBL-EB infections in humans and the burden of illness (BOI) have been published in the literature. The primary outcomes examined include mortality, length of hospital stay (LOS), treatment failure and health care costs [198,199]. 

Some studies in European populations have compared the severity of ESBL-producing infections caused by *E. coli* and *K. pneumoniae.* In a comparison of bacteremia caused by ESBL-*E. coli* and *K. pneumoniae,* there was a greater risk of organ failure in cases caused by *K. pneumoniae,* but no significant difference in mortality [200]. Poorer disease outcomes are also reported in cases of rectal colonization by ESBL-*K. pneumoniae* compared to ESBL-*E. coli* [201]. These studies suggest that ESBL-*K. pneumoniae* infections are associated with greater disease severity than ESBL-*E. coli* infections [200,201]. 

All-cause mortality and 30-day mortality are BOI measures which estimate the patient burden and have been identified as significantly higher in patients with ESBL-EB infections compared to patients with infections caused by non-ESBL-EB [194,202,203,204,205,206,207,208,209,210]. A systematic review and meta-analysis by Mackinnon et al. examining the BOI of resistant *E. coli* in human infections found an overall odds ratio (OR) of 2.27 when considering the impact of 3GC resistant *E. coli* on mortality, which was similar to studies by Rottier et al. and Schwaber & Carmel which identified an OR value of 2.35 and relative risk of 1.85, respectively [198,206,207]. This increased mortality in ESBL-EB infections has been associated with delayed appropriate antimicrobial therapy and an increased likelihood of treatment failure [209]. However, the systematic review by MacKinnon et al. [198] also identified infrequent studies which did not demonstrate an association or significant difference between ESBL-*E. coli* and non-ESBL-*E. coli* infections. Additionally, all-cause mortality does not consider confounding factors such as co-morbidity, thus the mortality may not be totally attributable to the 3GC resistant bacterium in question.

LOS and LOS post-infection are important BOI measures, which along with treatment failure and hospital costs are used to demonstrate the burden on the healthcare system. LOS post-infection is considered a better measure, as overall LOS can vary depending on the type of infection and can introduce a time dependent bias, including hospital stay duration not associated with the ESBL infection in question [211,212,213]. Several studies have demonstrated a significant increase in LOS due to ESBL-EB compared to non-ESBL-EB infections [194,203,209,214,215]. Compared to 3GC-susceptible infections, resistant bloodstream and respiratory tract infections require an increased duration of antimicrobial treatment and longer hospital stays [216]. The length of hospital stay may be influenced by ESBL-patients receiving inappropriate initial antimicrobial therapy and ESBL-patients requiring a longer treatment course of the therapy [215]. MacVane et al. [215] found only 23.6% of patients hospitalized with ESBL-*E. coli* and *Klebsiella* UTIs received appropriate initial antimicrobial therapy, compared to 98.2% of patients hospitalized with non-ESBL-*E. coli* and *Klebsiella* UTIs.

According to McDonald et al. [217], a treatment failure may be clinical (associated with persistent infections requiring therapy modifications or death of the patient) or microbiological (failure to eliminate the offending microorganism). In the literature examined, treatment failure was found to be more common among ESBL-EB infected patients than among the control groups. When considering ESBL-EB infections of all types, both MacVane and Maslikowska noted treatment failure differences of 56% (*p* < 0.001) and 14% (*p* = 0.03) between ESBL-EB and non-ESBL-EB groups, respectively [203,215]. Similarly, Tumbarello reported treatment failure rates which were three times higher among patients with ESBL-EB bloodstream infections after the first 72 h of therapy (OR = 3.21, *p* < 0.001) [209]. Esteve-Palau et al. [218] also demonstrated that clinical failure in the first seven days was associated with ESBL production and prior antimicrobial treatment. Conversely, Park et al. [219] did not demonstrate that ESBL production was significant when considering clinical and microbiological failure. The type of infection, the therapy duration and appropriate vs. in-appropriate initial therapy may be confounding factors and introduce bias in the results.

Hospital costs are higher overall for patients with ESBL-EB infections compared to patients with infections caused by non-ESBL-EB, largely due to a longer hospital stay and the location of the hospital stay (e.g., intensive care unit) [194,203,209,214,215]. In a USA cohort study, the patients hospitalized with ESBL-*E. coli* or *Klebsiella* spp. infections other than UTIs incurred a mean additional hospital cost of $27,387 CAN compared to matched patients hospitalized with non-ESBL-EB infections [2,214]. Finlay et al. [2] reported estimates of hospital costs incurred due to ESBL-producing UTIs as compared to susceptible UTIs ranging from $955 to $1556 CAN. They also estimated that a patient with a hospitalized ESBL infection in Canada will cost $20,893 CAN per patient, relative to a patient with no infection. Similarly, in a Spanish study, a significant association between ESBL production in UTIs and higher hospital costs was identified (*p* = 0.008) [218]. Finally, studies by Lee et al. [214] and Tumbarello et al. [209] demonstrated a 1.5–1.7 times increase of hospital costs for ESBL-patients when compared to non-ESBL-patients. Additional hospital costs have been directly related to an increased post-infection length of hospital stay, with ESBL-patients spending 7–9.7 days longer in the hospital [209,214]. A summary of examples from the literature demonstrating the BOI associated with ESBL-EB infections is included in Supplemental Appendix A**.**

#### 3.5.5. Summary of Data Quality and Level of Concern

The overall average data quality for this section is 6.2. The lack of Canadian BOI data concerning ESBL-EB is a major gap. Given the available information, it is reasonable to expect increases in mortality and hospital costs, encompassing increased LOS and treatment failures, from ESBL-EB, compared to infections caused by susceptible bacteria. The level of concern is estimated at three, based on these concerns.

### 3.6. Risk Management Information

#### 3.6.1. Identification of Risk Management Options to Control the AMR Hazard along the Production to Consumption Continuum

An AMR hazard may be introduced into the seafood to fork continuum at multiple points, and risk management options for hazard control were discussed in a recent risk profile by Loest et al. [15]. The following table summarizes risk management options in the seafood to fork continuum which have the objective of improving the microbiological safety of seafood and include elements to diminish the development, selection and mobilization of ARGs (Table 1). Although general in nature, they apply equally as well to ESBL-EB as to other AMR/ARG/bacteria combinations of concern. The principal risk management options considered target aquaculture production practices, industry support and regulation and quality assurance programs in the processing sector. 

Imported seafood may be raised under regulatory principles and industry support systems which differ from the Canadian context. Aspects such as drug approval and dosage, cultured species, siting of aquaculture operations, as well as availability of diagnostic services and fish health professionals may vary widely. The potential risk of AMR in imported seafood includes all these aspects of production including transport and handling, among others [220]. Predicting the risk may not be possible and may only be managed once the seafood has arrived in Canada. In these cases, risk management options to decrease the propagation of AMR in the domestic seafood to fork continuum must be explored.

#### 3.6.2. Utilization of WGS as a Surveillance-Based Risk Management Tool to Control the AMR Hazard along the Seafood to Fork Continuum

The identification of AMR food safety issues and antimicrobial hazards and defining the source and transmission pathways in the food chain are essential for the control and prevention of negative outcomes in the human population. The proximity of aquacultured products to the aquatic environment adds to the complicated context of AMR transmission and exemplifies the need to understand AMR generation and transmission along the One Health continuum. Until recently, these relationships have remained somewhat unknown due to the limitations of methods in describing the molecular environment of ARGs in bacterial isolates and the inherent imprecision in techniques used to identify the relationships between organisms with similar resistance patterns. WGS can provide information contributing to the understanding of the development or selection of ARGs and genetic factors (plasmids, transposons, etc.) which facilitate transmission and possible associated risks. Additionally, WGS can be critical in clinical applications, AMR surveillance programs and outbreak investigations.

#### 3.6.3. Source Attribution and Mitigation of ARGs and/or Resistant Strains and Utility of WGS

Important ARGs and AMR phenotypes have been identified in the aquatic environment [221,222]. These may have a genesis in terrestrial wastewater effluents including human and agricultural sources and aquaculture as a direct contributor to the aquatic environment [223]. Their implication in AMR development and transmission to humans through the food chain is complex and the quantification of their role is difficult, due to lack of information concerning bacterial transmission pathways, inter- and intra-species ARG exchange and co-selection and cross-resistance issues. The average quantity of medically important antimicrobials sold for use in animals in Canada in 2019 was 955,413 kg [224]. Although Canadian aquaculture represents only a small percentage (1.3%) of the total usage at approximately 12,507 kg [224], the impact of direct deposition of antimicrobials into the aquatic environment, the intricate aquatic microbiome/resistome, the effects of terrestrial effluent and post-harvest contamination at the processing and retail levels should also be considered.

WGS can be used as a surveillance tool to signal emergence or transmission between different microbiomes across humans, animals and the environment. The usefulness of WGS in establishing a relationship between the human and aquatic environment has already been demonstrated. Studies examining ARGs upstream and downstream of human anthropogenic sources (e.g., wastewater, city) have consistently shown an increase in ARG numbers and types, as well mobile genetic elements demonstrating the impact of human effluents on the aquatic environment [223,225]. Lee et al. [226] also noted that ARGs in the aquatic environment upstream of anthropogenic influence were quite different from those downstream. Interestingly, the metagenomic data used indicated that the human gut microbiota identified in the aquatic environment differed from the aquatic bacterial composition downstream, whereas the ARGs did not. This would indicate that the maintenance of ARGs in the aquatic environment is due to ARG proliferation and not human-sourced bacterial multiplication, demonstrating the importance of preventing bacterial contamination from human waste to avoid the introduction of ARGs into the environment. 

WGS can also help differentiate bacterial phenotypes and genotypes, investigating relatedness and ultimately source attribution. Runcharoen et al. [227] found that human clinical isolates and environmental bacterial strains of *E. coli* (ESBL and carbapenem resistance phenotypes) formed distinct genetic groupings. However, in a study comparing isolates from hospital sewage and the adjacent aquatic environment, three clinical-environmental pairs of *E. coli* isolates taken from hospitals and adjacent canals (within 0.5 to 5.2 km) were shown to be closely related (differed by ≤5 single nucleotide polymorphisms). In a study by Nadmipalli et al. [98], they demonstrated that *E. coli* with ESBL phenotypes were phylogenetically related among healthy colonized patients and isolates from meat and seafood.

The tracking of specific gene types or families of interest is also possible using WGS. Public health organizations can identify high-risk resistant clones regionally and globally and evaluate the geographic distribution [228]. Comparison of the whole DNA sequences of different isolates can help reconstruct transmission chains and trace outbreaks [229]. Mutations can arise in ARGs of importance to human health, which decrease bacterial susceptibility. WGS surveillance activities could be used to identify important genetic changes and permit the institution of adequate control measures. For example, the *bla*_TEM_ and *bla*_SHV_ genes were responsible for the majority of ESBL activity in the 1980s, whereas the *bla*_CTX-M_ genes encoding various CTM-X enzymes now predominate [230]. 

AMU is cited as the biggest driver of AMR in aquaculture [108]. This varies widely from country to country. For example, in a survey of four Asian countries, Rico et al. [24] identified 26 antibiotics being used. Conversely, in Canada, only three antimicrobials are approved for aquaculture (sulfadimethoxin-ormetoprim, Category II-High Importance to human medicine and oxytetracycline and florfenicol, Category III-Medium Importance) [4]. Here, co-selection or mobilization of ARGs from the aquatic resistome may be potentially of greater importance than resistance developed to the antimicrobials actually used. It is therefore important to understand the genetic environment of the ARG responsible. The presence of mobile genetic elements, including integrons, transposons and plasmids, in addition to accessory genes, e.g., pathogenicity genes, can be detected with WGS. With more advanced techniques (long read sequences), the reconstruction of plasmids is possible and the genetic context of ARGs with other horizontal gene transfer elements can be elucidated [231]. An ARG placement in the bacterial genome is important as it relates to a transfer risk, and its co-presence on a broad host range plasmid with other ARGs is more concerning than a gene found on the bacterial chromosome.

Plasmid types which are revealed by WGS analysis can have several implications in risk evaluation. The plasmids present are identified by recognizing the replicon (Rep) and mobility (Mob) type structures, which possess conserved replication and mobility functions and can be associated with particular ARGs. IncF plasmids, for example, have been associated with the global transmission of *bla*_CTM-X-15_ and can integrate a wide variety of ARGs [230,232]. However, identifying the same plasmid types in two strains may not indicate relatedness. Even with more resolution of plasmid content, there are still challenges as often there is repeated genetic material, more than one replicon type within the plasmid and possibly more than one plasmid in an isolate.

The use of WGS has some important limitations. To identify emergent or heretofore unrecognized ARG types, the phenotypic analysis of at least a percentage of isolates should be retained to help uncover those genes, where their role in resistance was not revealed using molecular techniques. Additional fundamental research would be needed to ascribe functionality to ARG candidates elucidated by WGS analysis. Certain WGS techniques may permit the identification of the location of the ARG in the isolate (chromosome, plasmid and association with integrin and transposon) but higher associated costs may limit their use [231]. The gene localization is important when evaluating transmissibility. Finally, although WGS analysis is increasingly available in highly developed countries, it remains difficult to implement in developing countries where resources and capacity are limited [233].

#### 3.6.4. AMR Surveillance Using Metagenomic Analysis and WGS

A metagenomic analysis refers to the untargeted replication and analysis of all DNA present in a sample using WGS techniques. From this, an idea of the taxonomic diversity, the presence and functionality of recognized genes, the discovery of novel genes and whole genome sequence recovery can be envisaged [234].

This technique offers certain advantages. It is important to note that the majority of bacteria from the aquatic environment are viable but nonculturable [235,236]. Additionally, the presence of non-pathogenic ARG carriers is often neglected. Using a metagenomic/WGS approach has an advantage of passing over the non-culturable hurdle, and ARG similarities can be deduced from sequence data giving an idea of the source: animal, human or environment. The resistome of the aquatic environment is an important gene repository and source of new ARGs and has been credited with the emergence of important ARGs, such as *qnr* for fluoroquinolone resistance, the mobile *bla*_FOX_ AmpC β-lactamase genes and the *bla*_CTX-M_ genes, respectively, which represent a risk for human health and may compromise treatment [237,238,239]. Therefore, WGS analysis could be used to monitor ARG circulation, identify ARG mutations of importance and predict future ARG emergence. 

This type of analysis has a few drawbacks. The identity of the bacteria harboring an ARG may be unknown, the activity of an ARG cannot be evaluated and its location in the genome of the bacterium may not be clear, whether chromosomal or plasmid-borne. The question of whether a living or dead bacterium and the potential posed for transmission would remain unanswered as this is a culture-independent technique. Additionally, metagenomic studies rely on gene libraries for comparisons of genetic reads that are generated. The absence of genes in the library limits the technique to genes that have already been identified or close relatives.

Aquaculture trade is a global industry and communication between countries or levels of investigation, including national, local public health authorities, experts from national reference laboratories and clinicians, can be challenging [240]. WGS can facilitate international investigations, lending clearer insight into ARG sources and transmission pathways, thereby suggesting avenues of containment and mitigation. WGS also provides a common language for the evaluation of ARG/AMR which is not hampered by differing scientific techniques and human interpretation of the data. As mentioned previously, WGS analysis is not universally available and support for developing countries, where the aquaculture industry exhibits the strongest growth, would be required. Increased surveillance of preoccupying seafood hazards, e.g., carbapenemases and ESBLs in shrimp, can identify and help resolve potential AMR food safety issues.

WGS methods can be utilized to examine each of the nodes in the seafood to fork continuum (Table 2). Combining knowledge of ARG sources and their genetic environment, transmission pathways will be better understood, leading to more informed decisions to limit the source transmission and enrichment of specific gene combinations. 

#### 3.6.5. Summary Data Quality and of Level of Concern 

The evaluation of risk management actions in the Canadian context are difficult at present, as a good proportion of the seafood production chain is located in exporting countries. Regulatory and enforcement differences between Canada and seafood exporters concerning the AMU can have an important impact on AMR in the final product. Auditing and approving production and processing practices of exporting countries, as outlined in the *Safe Food for Canadian Regulations* could ensure the safety of the seafood product, but may not address the presence of AMR. The absence of targeted surveillance programs capable of following prevalence trends of ESBL-EB bacteria in salmon and shrimp and the inability to evaluate outcomes of risk management decisions constitute a significant gap. 

### 3.7. Evaluation of Available Information and Major Knowledge Gaps

Information utilized for the development of this risk profile was evaluated for each section as to the quality of the data and the identification of data gaps (Appendix A). Information concerning the antimicrobial-resistant microorganisms and determinants associated with ESBLs was available and well represented, with the exception of that concerning the distribution, frequency and concentrations in the seafood to fork continuum. The lack of ongoing surveillance activities was a major source of uncertainty, as well as a lack of knowledge concerning AMU practices and AMR in countries exporting seafood products to Canada. Baseline surveillance information is necessary to conduct a hazard and exposure assessment, both initial steps of a foodborne AMR risk analysis. Good Canadian information was available when examining human uses of the antimicrobial agent, and uses of antimicrobials in domestic aquaculture. However, although human BOI attributable to ESBLs is well documented in the peer-reviewed literature, information specific to Canada was lacking. 

## 4. Discussion

This risk profile presents information permitting the characterization of the presence of ESBL-EB in Canadian seafood as an AMR food safety issue. Data concerning the distribution and frequency of the AMR hazard in the Canadian seafood chain are lacking, making a formal risk analysis, a quantitative microbial risk assessment and the implementation of constructive risk management options challenging at this time. The implementation of surveillance of AMR in Canadian retail seafood is necessary, whether targeted surveillance or pilot studies, to better inform and guide a risk analysis process.

### 4.1. The Importance of This Risk Profile

AMR has far-reaching socio-economic and health impacts in Canada. It has been estimated that in 2018, AMR reduced the Canadian gross domestic product by 2 billion dollars, cost the Canadian health care system 1.4 billion dollars and was responsible for 14,000 deaths [2]. Resistance to β-lactams caused by the presence of ESBL-EB is increasing in Canada healthcare institutions [165,169]. In Canadian hospitals between 2007 and 2018, the proportion of ESBL-*E. coli* and ESBL-*K. pneumoniae* isolates increased from 3.3 to 11.2% [169]. This potentially limits treatment options and engenders an increased reliance on broad spectrum antibiotic classes, such as carbapenems, considered as a last resort therapy for multidrug-resistant or extensively drug-resistant pathogens.

Shrimp and salmon are the two principal seafood products consumed in Canada and are mainly aquacultured organisms, salmon domestically and shrimp as an imported product. In Canadian aquaculture, only three antimicrobials are authorized for use, including oxytetracycline, ormetoprim-sulfadimethoxine and florfenicol. Regulations concerning the sale of antimicrobials are governed at the federal level by Health Canada and the Canadian Food Inspection Agency via the *Food and Drugs Act* and its regulations, whereas AMU is regulated at the provincial/territorial level [241]. Additionally, Fisheries and Oceans Canada requires the reporting of AMU for marine and freshwater aquaculture [242]. Similar regulatory structures may not be present in all countries providing seafood products to the Canadian retail market [243,244,245]. A regulatory divergence between domestic and imported aquaculture products could represent a different likelihood for AMR exposure. Although 3GC use in aquaculture has not been reported, multiple classes of antimicrobials including aminopenicillins and first-generation cephalosporins (cephalexin) were used in some countries [24,108]. Canada’s dependence on importations of popular seafood types such as shrimp was an additional motivator for the development of this risk profile.

### 4.2. Risk Profile Preparation Using the Codex Recommendations

Similar to previous publications, this risk profile was developed following recommendations by Codex, as one of the first steps in a foodborne AMR risk analysis process [13,14,15]. This risk profile differed from previous risk profile publications in that the “hazard of concern” examined included more than one bacterial genus and/or species, the Enterobacterales order. A greater emphasis was also placed on genes, rather than bacterial species. The presence of ESBLs in seafood was often a highlight in pertinent peer-reviewed publications rather than a particular bacterial species. This demonstrates the increasing recognition of the importance of the ESBL gene or determinant and its inter/intra-species mobility, and the role played by One Health interactions of human, environment and food animals. 

As this risk profile focused more on ARG transmission, a discussion of molecular techniques was warranted. Molecular techniques, particularly WGS, were presented as an element for consideration in this risk profile as a risk management tool. With the rapid advance of sequencing technologies and decreasing costs, WGS is becoming a standard in AMR surveillance. WGS provides analytical granularity which allows for phylogenetic comparisons, source attribution studies, detection of AMR genetic diversion and the capacity to pursue a One Health understanding of AMR. Accurate and rapid diagnostics and genetic ARG evaluation could also be helpful when dealing with fastidious pathogens in the clinical setting.

### 4.3. Data Gaps Identified

Unlike a previously published risk profile examining carbapenem resistance in *E. coli* in retail salmon and shrimp, ESBL-EB have received more attention in the published literature [15]. However, there is still a lack of Canadian targeted surveillance information with the exception of a communication from Janecko (N. Janecko, personal communication, 2022). The majority of the collated data was from other geographical regions identified, similar to Loest’s study. Although not a data quality issue, there was also little to no information concerning Enterobacterales/ESBL prevalence in salmon or salmon products. Studies examining prevalence in the salmon-specific food-to-fork continuum (production, harvest and processing) were conspicuously absent. Additionally, as previously noted, importation represents an important source for Canadian retail salmon, which could affect ESBL-EB prevalence. Mughini-Gras et al. [246] found ESBL and plasmid-encoded AmpC positive *E. coli* prevalent in seafood in the Netherlands. This raises concerns from the viewpoint of a population which consumes raw, undercooked cured or smoked seafood, which mirrors the Canadian situation. Furthermore, importation is an important source of seafood, and there is scant information available concerning the prevalence of ESBL-EB in seafood, and AMR is prevalent in the aquatic environment [246].The lack of information concerning the prevalence of ESBL-EB in Canadian retail seafood, particularly in salmon and shrimp highlights a need for increased surveillance, which would facilitate future and updated hazard/risk evaluation. Finally, as noted previously, studies examining human BOI indicators such as mortality, hospital costs, LOS and treatment failures have been described in the literature for ESBL-EB, but not in the Canadian context. This information is important for a risk analysis and represents a major data gap.

### 4.4. Surveillance Considerations

As described previously, a recent study of Canadian retail seafood identified ESBLs in imported shrimp. In this study which covered a sampling period from 2014–2016, an overall prevalence of ESBL-EB of 4.2% (52/1231) in fish and seafood where shrimp were the source of over half (63%, 33/52) of all ESBL-EB isolates identified (N. Janecko, personal communication, 2022). This study also demonstrated a predominance of the *bla*_CTX-M-15_ gene, which is also the principal ESBL-encoding ARG identified in *E. coli* and *K. pneumoniae* in Canadian hospitals [169]. The study by Janecko appeared to demonstrate a predominance of ESBL isolates from imported seafood; however, it should be noted that only 85 fish and shrimp samples were of Canadian-origin among the 1231 seafood samples. An adequate sample of domestic seafood should also be included in future Canadian surveillance initiatives. The increased sensitivity of genetic methods as previously described could also be used as an adjunct to standard culture methods to improve detection of bacteria and ESBLs in seafood.

Important ESBL determinants have been shown to be often carried on mobile genetic elements such as plasmids with other ARGs, and are subject to selection or co-selection when subjected to AMU in aquaculture or environmental contamination [77,113]. Therefore, evaluation of ARGs and their genetic structure is an important element to consider in future surveillance programs. The use of WGS techniques such as long read sequencing could help localize ARGs and define their genetic environment, which has an important impact when considering an ARG hazard, a food safety issue or ARG source attribution [231,247]. If the ARG presence is the hazard in question, and not the harboring bacteria, a metagenomics approach could be considered. Duarte et al. [248] demonstrated the possibility of using metagenomics to identify country-specific ARG markers useful for source attribution at the country level, as well as between animal and human resistomes.

The frequency of detection of bacteria (i.e., apparent prevalence) in seafood by standard culture practices has the potential to underestimate human exposure. For example, Wang et al. [249] demonstrated *Salmonella*/*Shigella* by PCR in both salmon and shrimp sampled in retail grocery stores, 14.3%/36.5% in salmon and 21.1/28.9% in shrimp and not by culture. Similarly, Shabarinath et al. [250] noted a *Salmonella* prevalence of 19% in retail shrimp using traditional culture methods as opposed to 59% with PCR. This discrepancy underlines the importance of the methodology used in prevalence studies and the comparability between studies, as well as the advantage of incorporating molecular techniques into surveillance systems. 

There is some discrepancy between WGS/predicted AMR and phenotypic AMR. Zwe et al. [251] found in their study of *Salmonella* in raw chicken that WGS missed hetero-resistant *Salmonella* populations which were detected using phenotypic techniques. However, other authors note good concordance between WGS predicted phenotypes and antimicrobial susceptibility testing phenotype results for certain bacterial species [252]. In a report from the European Committee on Antimicrobial Susceptibility Testing, a recommendation for comparisons using epidemiological cut-offs (differentiating between wild-type and non-wild-type bacterial species) is recommended, as predicted phenotypes may not correlate well with clinical breakpoints [253]. 

### 4.5. Where Do We Go from Here to Evaluate Risk?

Foodborne AMR risk assessment is composed of four principal elements: hazard identification, exposure assessment, hazard characterization and risk characterization. This risk profile provides major information necessary to characterize the foodborne AMR hazard, including important genotypes and seafood types and origins which are of concern. An exposure assessment and the elucidation of an exposure pathway would be possible from the Canadian perspective where domestic aquaculture production parameters could be ascertained, such as preharvest selective factors (AMU), dissemination factors (fish waste, and processing) and human exposure (concentration of ARGs of concern in retail seafood and transmissibility to humans). Most of these elements of the exposure pathway would require an active surveillance system to be in place. For imported seafood, access to information regarding the exposure pathway of an AMR hazard would be unavailable, and an unrestricted/inherent risk analysis could be used which assumes there were no safeguards upstream of the retail level and where conditions of production are unknown [220]. Here again, surveillance is needed to obtain baseline data from which the risk analysis process can be undertaken.

Leveraging inspection and surveillance programs currently in place could be envisioned at importation and retail levels (the Canadian Food Inspection Agency, CIPARS, etc.). Although current Canadian inspection procedures for imported and domestic seafood are focused on the presence or absence of particular types and numbers of pathogenic organisms and not on ARGs, these samples, which are already collected, could be submitted for bacteriology and susceptibility testing [163]. Several retail seafood sampling pilot projects have been undertaken by CIPARS between 2008 and 2016 to assess AMR in seafood (N. Janecko, personal communication, 2022). CIPARS already participates in the collection of retail food samples of several species across Canada, to assess AMR, and the collection of retail seafood samples is planned to begin in the near future. Based on the findings of this risk profile, there is not enough information to classify ESBLs in seafood as a “risk” to Canadians. The information obtained from future seafood surveillance activities by CIPARS will be needed to assess this AMR food safety issue/hazard. Surveillance activities should include domestic and imported species; however, the evidence to date suggests a higher prevalence of ARG hazards in imported seafood such as shrimp in Canada. Moreover, the European Union has developed AMR surveillance programs which take this into account [254]. WGS techniques should be utilized to answer questions concerning source attribution, the likelihood and frequency of horizontal gene transfer (co-resistance) and the identification of important genotypes and/or genetic structures. 

Legislation concerning sampling of the aquatic environment around aquaculture sites exists in Canada [255]. Additionally, a Canadian project in development guided by the National Microbiology Laboratory, AMRnet, aims to gather and analyze antimicrobial susceptibility testing data from private laboratories, which could include those working with aquaculture species [256]. In a holistic or One Health view of AMR, consideration could be given to incorporating these existing structures into a surveillance suite for aquaculture in Canada.

Similar to a previous risk profile examining seafood, the Codex Guidelines proved flexible enough to evaluate a complex AMR food safety issue, in this case, multiple bacterial species, and multiple environments of the seafood chain, with an emphasis placed on ARGs responsible for resistance. The examination of similar commodities, with a similar risk environment in the article published by Loest et al. [15] permitted a streamlining of the information presented, saving time and labor in the production this manuscript.

While the information contained in this risk profile was not substantial enough to conclude whether or not there is a human health risk related to ESBLs in Enterobacterales in salmon and shrimp available for consumption by Canadians, ESBLs in imported seafood available at the retail level in Canada have been identified. In Janecko’s study, the majority of the ESBL genes identified were those encoding CTX-M-15, the most prominent gene types identified in Canadian hospital surveillance programs, which raises questions concerning source attribution (N. Janecko, personal communication, 2022) [169].

To better understand the risks associated with this AMR food safety issue and address the principal data gaps mentioned previously, AMR surveillance needs to be developed and maintained. Salmon and shrimp could be used in initial surveillance activities, representing domestic and imported products. To comprehend the flux of ESBL ARGs, as part of an exposure assessment and to investigate source attrition, WGS should also be incorporated into surveillance activities.

## Figures and Tables

**Table 1 antibiotics-12-01412-t001:** Summary of risk management options.

Measures to Reduce the Risk Related to the Selection and Dissemination of Foodborne AMR Microorganisms			
Goal	Risk Management Option	Action	Effect
Decreased the need for AMU	optimal culture methods (e.g., biosecurity, vaccination, localization of site, stocking density, fallowing, water quality, nutrition)	reduce (↓) stress ↓ contamination ↓ antimicrobial accumulation in the environment ↓ terrestrial contamination improved water quality	↓ disease prevalence ↓ AMU ↓ selective pressure on resistome
	optimize industry support	ensure availability of diagnostic services and extension support ensure availability of approved medication/doses for aquatic species - support research, development - regulatory approval of alternatives (e.g., vaccines, non-chemical)	accurate diagnosis optimal disease management AMU stewardship
	adequate regulatory environment, adoption of prudent use guidelines	establishment and enforcement of AMU practices in aquaculture	AMU stewardship
**Measures to minimize the contamination and cross-contamination of food by AMR microorganisms**			
Decreased propagation of AMR in the seafood to fork continuum	HAACP plans in processing and distribution chain importation regulations and inspections (e.g., Safe Food for Canadians Regulations) development of surveillance programs for imported and domestic seafood	cold chain maintenance decrease or avoid contamination/cross contamination micro-organism control chlorination/ozonation/irradiation hygiene thermal inactivation monitoring of effectiveness block-chain, farm-to-fork tracking	↓ bacterial multiplication ↓ contamination ↓ exposure of consumer to AMR

**Table 2 antibiotics-12-01412-t002:** Proposed WGS applications for the seafood to fork nodes.

Seafood to Fork Nodes	WGS Applications
Aquatic environment	- investigation of the aquatic resistome - ARG circulation, mutation and co-carriage
Aquaculture	- AMU reduction/↓ selective and co-selective pressure - clinical applications, e.g., diagnosis, therapy - identification of vaccine candidates - biosecurity, e.g., screening of broodstock for infectious diseases, preventing vertical transmission of disease
Harvest/Transportation	- source attribution, e.g., new isolates/genes are being introduced by people to the aquaculture products during harvest or transport
Processing	- typing and epidemiologic/phylogenetic investigations to identify critical control points during processing, including occupational inputs - identifying preventative measures (e.g., HAACP)
Retail	- continuity of epidemiological investigations, closer to outbreaks of human disease - similar to processing - surveillance - evaluation of human exposure to AMR via consumption of seafood products

## Data Availability

All data generated or analyzed during this study are included in this published article (and its Appendix A).

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
