# Peer review of "Extended Spectrum β-Lactamase-Producing Enterobacterales of Shrimp and Salmon Available for Purchase by Consumers in Canada—A Risk Profile Using the Codex Framework"

_antibiotics, 2023, doi:10.3390/antibiotics12091412_

Round 1
Reviewer 1 Report
Uhland et al present their review on risk profiling of ESBL-producing Enterobacteriaceae in shrimp and salmon ready for purchase and subsequent consumption in Canada. The review principally draws from the principles of One Health and seems novel. It also has a high epidemiological significance, as aquatic ecosystems are important reservoirs for antimicrobial-resistant bacteria. In addition, the authors have provided sufficient background, making the importance of their study apparent. Although the authors have explained the approach to the review, the details of how the literature search was done are unclear. Besides, there are some portions of the manuscript which though captured as part of the Results, need to be in the Introduction or Methodology. There are also some issues concerning inconsistent referencing and a few minor grammatical errors, specific comments of which are as follows:
a. Issues with Referencing
The authors need to be consistent in their use of Vancouver format for their citations. Examples of portions where this issue needs to be addressed include Lines 47, 59, 81, and 87.
b. Misplaced Contents
The contents of Lines 102 to 108 belong in the Methodology.
The contents of Lines 109 to 110 belong in the Introduction.
c. Issues with Grammar
Line 32: Please hyphenate “far reaching”.
Lines 34 to 36: The communication here seems conflicting. Is the estimate specific to the year 2018 or for each year?
Line 39: Please hyphenate “3rd generation”.
Line 40: Please hyphenate “third generation”.
Line 42: Please introduce a comma after the “(WHO)” and delete the “the” that immediately follows it.
Line 44: Please hyphenate “food producing”.
Line 48: Please write “product” in plural form.
Lines 54 to 55: Please rewrite “principle” as “principal”.
Line 81: Please hyphenate “peer reviewed”.
Line 91: Please rewrite “summed” as “summed up”.
Line 152: Please hyphenate “seafood to fork”.
Line 169: Please replace the first “have” with “are”.
Minor edits are needed.
Author Response
Response to Reviewer 1
Manuscript ID: antibiotics-2566163, entitled “Extended spectrum β-lactamase producing Enterobacterales of shrimp and salmon available for purchase by consumers in Canada - A risk profile using the Codex framework”
I would like to thank you for the time dedicated to the review of our manuscript. The comments and suggestions for changes were also appreciated, and I believe they improve the paper content, improving clarity and flow. Our responses to your comments are noted below in italicized text, and requested changes in the manuscript are indicated by manuscript line and with a highlighted red font.
Reviewer 1 :
Comments and Suggestions for Authors
Uhland et al present their review on risk profiling of ESBL-producing Enterobacteriaceae in shrimp and salmon ready for purchase and subsequent consumption in Canada. The review principally draws from the principles of One Health and seems novel. It also has a high epidemiological significance, as aquatic ecosystems are important reservoirs for antimicrobial-resistant bacteria. In addition, the authors have provided sufficient background, making the importance of their study apparent. Although the authors have explained the approach to the review, the details of how the literature search was done are unclear.
Thank you for this comment. The risk profile as described by Codex does not follow a rigid literature search outline as that described for scoping and systematic reviews, but as an information gathering tool to guide future actions or risk assessments.
CODEX describes a risk profile thusly; a “… risk profile presents, in a concise form, the current state of knowledge related to the food safety issue, describes current control measures and RMOs that have been identified to date and the food safety policy context that will influence further possible actions. It is important to note that the risk profile is a scoping exercise to describe and define the pertinent factors that may influence the risk posed by the hazard. It is not intended to be an abbreviated version of a risk assessment”.
We have added the following text to the methods section to clarify the approach we used to identify the pertinent literature: (lines 96-102), Peer-reviewed literature was identified by performing an exhaustive search via the internet using multiple web browsers and publicly-available databases including; PubMed, PubMed Central, Web of Science, Scopus, and Google Scholar. Various key words were selected on the basis of expert opinion and the need of this specific risk profile, such as Enterobacterales, Enterbacteriaceae and the related individual species, seafood, salmon, shrimp, aquaculture, ESBL and specific ESBLs, antimicrobials, specific drugs and/or resistance.
Besides, there are some portions of the manuscript which though captured as part of the Results, need to be in the Introduction or Methodology. There are also some issues concerning inconsistent referencing and a few minor grammatical errors, specific comments of which are as follows:
These are addressed in the specific comments below. Of note, we have checked the entire manuscript including consistency in referencing and error corrections.
- Issues with Referencing
The authors need to be consistent in their use of Vancouver format for their citations. Examples of portions where this issue needs to be addressed include Lines 47, 59, 81, and 87.
Thank your for the suggestion. The text was reviewed and corrections were made to those examples cited, as well as others if identified.
- Misplaced Contents
The contents of Lines 102 to 108 belong in the Methodology.
Agreed. Although we feel it is important to retain a part of this phrase when speaking to the “Description of the Food Safety Issue”, a phrase of introduction of the matter in question in the methods makes it easier to follow. The following text was moved/added to the Methodology: (lines 85-89), The AMR food safety issue explored in this risk profile, is the presence of ESBLs in Enterobacterales in salmon and shrimp available for purchase by consumers in Canada. Unlike the other two Canadian AMR risk profiles previously published using the Codex Guidelines [14,15], this risk profile focused on a resistance pattern expressed by one or more genes for multiple bacterial species.
The contents of Lines 109 to 110 belong in the Introduction.
This idea was expressed also in the Introduction (line 44-46).
- Issues with Grammar
Line 32: Please hyphenate “far reaching”.
Done as suggested.
Lines 34 to 36: The communication here seems conflicting. Is the estimate specific to the year 2018 or for each year?
Thank you. The words “per year” were removed to make the phraseology clearer.
Line 39: Please hyphenate “3rd generation”.
Done as suggested , changed to “third-generation" (in entire manuscript)..
Line 40: Please hyphenate “third generation”.
Done as responded above.
Line 42: Please introduce a comma after the “(WHO)” and delete the “the” that immediately follows it.
Done.
Line 44: Please hyphenate “food producing”.
Done.
Line 48: Please write “product” in plural form.
Done.
Lines 54 to 55: Please rewrite “principle” as “principal”.
Done.
Line 81: Please hyphenate “peer reviewed”.
Done.
Line 91: Please rewrite “summed” as “summed up”.
Done.
Line 152: Please hyphenate “seafood to fork”.
Done.
Line 169: Please replace the first “have” with “are”.
Done.
Reviewer 2 Report
From my point of view, the manuscript can be published without any modifications. The subject is of great interest and actual. Taking into consideration that the manuscript is a review, it is very well structured and documented with a large number of topical bibliographical references and supplementary material. I congratulate the authors.
Author Response
Response to Reviewer 2
Manuscript ID: antibiotics-2566163, entitled “Extended spectrum β-lactamase producing Enterobacterales of shrimp and salmon available for purchase by consumers in Canada - A risk profile using the Codex framework”
I would like to thank you for the time dedicated to the review of our manuscript. We appreciate your understanding of our manuscript and recognition of the authors’ efforts.
Reviewer 3 Report
Please see the attached file for more detailed comments and suggestions

Author Response
Please see attachment. As I was examining cited line numbers and reported changes, I noted an omission. Please accept my apologies for the confusion. This version should be used. Thank you.

Reviewer 4 Report
This article describe the current 70 state of knowledge of ESBL-EB in relation to Canadian retail shrimp and salmon, includ- 71 ing factors that may influence the potential risk posed, current and proposed risk man- 72 agement strategies, and the relevant policy context. The second was to identify subsequent 73 risk analysis steps to provide direction for policy makers concerning risk management 74 decisions. While going through this paper, i have following suggestions
Q:01- Extended spectrum beta lactamases producing enterobacterales are only resistant to 3rd generation cephalosporins? Why not resistant to penicillin and other cephalosporin generation?
Q:02- What could be the reason of presence of (ESBL-EB) in aquatic environment only?
Q:03- ESBL-EB are considered as human pathogens what kind of infections they cause?
Q:04- ESBL-EB includes a list of pathogens, mention names of any two or three related pathogens as example.
Q:05- Consumption of salmon and shrimp is only seal food containing ESBL-EB, no other aquatic food contain ESBL-EB?
Q:06- During processing ESBL-EB contamination can be overcome by alternation in temperature and pH?
Q:07- Antimicrobial agents such as macrolide, quinolones, sulfonamide have also been detected in aquatic environment then why ESBL-EB selected in study?
Q:08- How can you justify that all the salmon and shrimps contains ESBL-EB? Mention samples for PCR ? (of Salmon and shrimp)
Q:09- What could be the relationship of seasonal difference of ESBL-EB carriage and patient’s infection?
Q:10- The principle risk management options considered target production practice, industry support and quality assurance in processing section could be helpful in eradicating ESBL-EB from food chain?
Some minor changes required
Author Response
Response to Reviewer 4
Manuscript ID: antibiotics-2566163, entitled “Extended spectrum β-lactamase producing Enterobacterales of shrimp and salmon available for purchase by consumers in Canada - A risk profile using the Codex framework”
I would like to thank you for the time dedicated to the review of our manuscript. The comments and suggestions for changes were also appreciated. Our responses to your comments are noted below in italicized text, and requested changes in the manuscript are indicated by manuscript line and with a highlighted red font.
Reviewer 4:
Comments and Suggestions for Authors
This article describe the current state of knowledge of ESBL-EB in relation to Canadian retail shrimp and salmon, including factors that may influence the potential risk posed, current and proposed risk management strategies, and the relevant policy context. The second was to identify subsequent risk analysis steps to provide direction for policy makers concerning risk management decisions. While going through this paper, I have following suggestions
Appreciate the reviewer’s insights.
Q:01- Extended spectrum beta lactamases producing enterobacterales are only resistant to 3rd generation cephalosporins? Why not resistant to penicillin and other cephalosporin generation?
Thank you for the questions. ESBL Enterobacterales typically demonstrate resistance to all penicillins, 3GCs and monabactams as described in section “3.2.2.2. Cross-resistance and/or co-resistance to other antimicrobial agents”. We had chosen the ESBL/3GC in Enterobacterales because of the importance attributed to these molecules.
As noted in the text, the 3GCs are of particular interest because of their importance in treating severe disease in humans. (lines 151-156) The antimicrobials under scrutiny for this risk profile, are the 3GCs, which are classified by the WHO as ‘Highest Priority Critically Important Antimicrobials’ and by Health Canada as ‘Category I - Antimicrobials of Very High Importance’[4,26]. The 3GCs are used to treat serious infections caused by multidrug-resistant Enterobacterales for which few treatment alternatives exist, and such infections may result from transmission of Enterobacterales, including E. coli, from non-human sources [26,27].
Q:02- What could be the reason of presence of (ESBL-EB) in aquatic environment only?
The reasons most often cited for the presence of ESBL-EB in the aquatic environment is in relation to human activity such as hospital and sewage effluents. These are noted in the text at lines 286-290.
Q:03- ESBL-EB are considered as human pathogens what kind of infections they cause?
The various infections caused by ESBL-EB in humans are addressed in the section “Information on adverse public health effects” (lines 680-734). Disease conditions described include; urinary tract infections (UTI), intra-abdominal infections and pneumonia among others.
Q:04- ESBL-EB includes a list of pathogens, mention names of any two or three related pathogens as example.
Thank you for the suggestion. Specific pathogens were added in the introduction to help situate the reader. (lines 39-42) The extended-spectrum β-lactamase (ESBL) producing-Enterobacterales (ESBL-EB) are a group of organisms which demonstrate resistance to third-generation cephalosporins (3GCs) and encompass several important human pathogens such as Escherichia coli, Salmonella spp. and Klebsiella pneumoniae.
Q:05- Consumption of salmon and shrimp is only seafood containing ESBL-EB, no other aquatic food contain ESBL-EB?
Good question. Although other seafood have been found to contain ESBL-EB, shrimp and salmon are the most commonly consumed seafood products in Canada, representing both imported and domestic seafood products. As such, they were chosen as to be examined using this risk profile.
Q:06- During processing ESBL-EB contamination can be overcome by alternation in temperature and pH?
Thank you for the question. It is certain that cooking will inactivate/eliminate ESBL-EB, as long as cross-contamination with raw products or from poor hygiene does not occur in processing. Organic acids have been used to prevent the growth of spoilage bacteria, in seafood products or fish meal, but I am unaware of their use to prevent bacterial contamination in seafood.
Q:07- Antimicrobial agents such as macrolide, quinolones, sulfonamide have also been detected in aquatic environment then why ESBL-EB selected in study?
Good question. The ESBL-EB were chosen in this study because of their detection in a pilot Canadian seafood surveillance study within CIPARS and because of their importance for human health.
Q:08- How can you justify that all the salmon and shrimps contains ESBL-EB? Mention samples for PCR ? (of Salmon and shrimp)
I’m unsure to which statement you refer. I don’t believe we intimated that all salmon and shrimp samples contain ESBL-EB, and certainly if this was implied by the text, you are absolutely correct to underline this and it should be corrected. Again, shrimp and salmon were chosen because of their preponderance in the Canadian consumption trends and a higher exposure to the population. I am also unsure of the meaning of the followup PCR question. Thank you.
Q:09- What could be the relationship of seasonal difference of ESBL-EB carriage and patient’s infection?
Interesting question. As noted in the text, in cited studies, there have been variations noted where colonization may be higher in the summer months and infection higher in the winter months. The colonization may be related to festival factors/activities (swimming, barbecues, pettings zoos etc.) whereas increased infections and antimicrobial use in the winter months were responsible for higher numbers of ESBL+ isolates. Hope our manuscript will draw additional investigations to further understand the relationship.
Q:10- The principle risk management options considered target production practice, industry support and quality assurance in processing section could be helpful in eradicating ESBL-EB from food chain?
The risk management options are useful to decrease and control the risk, but not eliminate it (as antimicrobial resistance or ESBLs are mother nature we may only minimize them). To evaluate the risk, and identify management options and evaluate their effectiveness, it is necessary to establish the frequency of the antimicrobial resistance hazard (ESBL-EB) in seafood. One of the major conclusions of this risk profile is need for the establishment of a seafood surveillance program to accomplish this.
Round 2
Reviewer 1 Report
The authors have significantly revised the manuscript, and it has improved compared to the previous submission.
Minor edits may be needed.
Reviewer 3 Report
Thank you for the helpful and thoughtful responses. The information you provided is very useful.
Please take a look and correct any typos, grammar issues, or formatting problems in the content.
I don't have any other feedback.
Reviewer 4 Report
I am satisfied with revised version and comments of authors